



# Experimental techniques for the calibration of lidar depolarization channels in EARLINET

Livio Belegante[1], Juan Antonio Bravo-Aranda[2,3], Volker Freudenthaler[4], Doina Nicolae[1], Anca Nemuc[1], Lucas Alados-Arboledas[2,3], Aldo Amodeo[5], Gelsomina Pappalardo[5], Giusseppe D'Amico[5], Ronny Engelmann[6], Holger Baars[6], Ulla Wandinger[6], Alexandros Papayannis[7], Panos Kokkalis[7], and Sergio N. Pereira[8]

[1]National Institute of Research and Development for Optoelectronics, 409 Atomistilor Str, Magurele, Romania
[2]Andalusian Institute for Earth System Research, Granada, Spain
[3]Department of Applied Physcis, Unversity of Granada, Granada, Spain
[4]Meteorological Institute, Ludwig-Maximilians-Universitat, Theresienstr. 37, 80333 Munich, Germany
[5]Istituto di Metodologie per l'Analisi Ambientale CNR-IMAA, C.da S. Loja, Tito Scalo, Potenza 85050, Italy
[6]Leibniz Institute for Tropospheric Research, Permoserstr. 15, 04318 Leipzig, Germany
[7]National Technical University of Athens (NTUA), Physics Department, Heroon ytechniou 9, 15780 Zografou, Athens, Greece
[8]Evora Geophysics Center, Rua Romao Ramalho 59, 7000, Evora, Portugal

*Correspondence to:* J. A. Bravo-Aranda (jabravo@ugr.es)

**Abstract.** Particle depolarization ratio retrieved from lidar measurements are commonly used for aerosol typing studies, microphysical inversion, or mass concentration retrievals. The particle depolarization ratio is one of the primary parameters that can differentiate several major aerosol components, but only if the measurements are accurate enough. The uncertainties related to the retrieval of particle depolarization ratios are the main factor in determining the accuracy of the derived parameters in such

studies. This paper presents different depolarization calibration procedures used to improve the quality of the depolarization data. The results illustrate a significant improvement of the depolarization lidar products for all the selected lidar stations that had implemented depolarization calibration procedures. The calibrated volume and particle depolarization profiles at 532 nm show values that fall within a range of values that are generally accepted in the literature.

## 1 Introduction

Uncertainties related to the influence of anthropogenic activities on the Earth's energy budget and climate change have lead to a real interest regarding the aerosols direct and indirect radiative effects. Measurements of vertically resolved aerosol optical properties (as the ones performed by lidar systems) try to reduce these uncertainties. These systems are laser-based instruments able to provide quantitative information on aerosol layering and their properties (Measures et al., 1992). The principle is based on the detection of backscattered light that results from the interaction of the emitted laser light with the atmospheric con-

stituents. Fig.1 shows the main components of a lidar system with polarising capabilities.. The emitted laser light is oriented towards the atmosphere by means of the emission optics. After the emitted light interacts with atmospheric constituents, the backscattered light is collected by a telescope and directed to the wavelength separation unit (WSU- named also the receiving





optics unit for this study), Polarizing Beam Splitter (PBS) and photomultipliers (PMTs). The receiving optics (mirrors, lenses and dichroic filters), the PBS, and the PMTs will be treated as distinct units, since the effects of each unit alters the depolarization profiles from different perspectives. The collected laser light contains information about to the optical properties of the atmospheric components, and consequently to their size, shape and composition. Methods to retrieve these properties from elastic backscatter, Raman, multi-wavelength and depolarization lidars are already described in details in the literature (Fernald

et al., 1972; Klett et al., 1981, 1985; Kovalev and Eichinger, 2004). According to their application, lidar systems have different configurations, channel combinations and geometries. For atmospheric studies, the configuration of a lidar system narrows down to several types and optical layouts.

A major breakthrough in atmospheric studies is the development of global lidar networks, able to provide systematic lidar dataflow with a large temporal and spatial coverage (Earlinet, 2014a). The European Aerosol Research Lidar Network - EAR-

LINET data (Pappalardo et al., 2012, 2014) is relevant for climatology, regional and large scale model assessment but also for special events such as Saharan dust outbreaks, transport of smoke plumes or volcanic ash over Europe (Earlinet, 2014b, d, e), (Papayannis et al., 2008; Ansmann et al., 2009; Ansmann and Bosenberg, 2003; Nicolae et al., 2013; Mona et al., 2012; Timofte et al., 2015; Mortier et al., 2013). The multi-wavelength depolarization Raman lidar systems used in EARLINET (3 backscatter + 2 Raman + 1 depolarization: $3\beta+2\alpha+\delta$ lidar systems - Mona et al., 2012), are capable to provide an extended set of optical

parameters for aerosol characterization, by assuring the quality of the products through internal data quality procedures. For depolarization studies, most of the lidar systems are designed to measure with two channels (parallel and cross).

Recent atmospheric studies that use remote sensing data have been dedicated to aerosol typing, microphysical inversion and aerosol mass concentration retrievals. Since all relevant parameters are shape dependent (Hervo et al., 2012; Hogan et al., 2012; Gross et al., 2015), the depolarization products obtained from lidar measurements proved to be essential, giving

the opportunity to distinguish between rather spherical particles with low depolarization ratios, and non-spherical particles with higher depolarization ratios (Gasteiger et al., 2014). Lidar measurements of particle linear depolarization ratio are often used to discriminate between low depolarizing (e.g. local aerosol) and high depolarizing aerosols (e.g. dust), or liquid and ice clouds, requiring only a relative measure of these parameters. At the present state, the uncertainties of these products are high for EARLINET lidars and any aerosol classification based on relative lidar depolarization profiles is challenging. For

aerosol typing and mass concentration studies, absolute values of particle linear depolarization ratio are needed. According to (Petzold et al., 2010; Gross et al., 2013; Burton et al., 2012), the particle linear depolarization values characterizing several aerosol species (or mixtures of aerosols) ranges around close values: for pure dust, the particle depolarization value at 532nm ranges from 0.30 to 0.39 and for dust mixtures from 0.1 to 0.30. The same issue emerges when discriminating between biomass burning aerosol mixed with mineral dust and industrial pollution aerosol, with values around 0.1 to 0.2 for the first and 0.04 to

0.1 for the second. Therefore, in order to discriminate between different types of particle, the uncertainty of the depolarization products must be reduced. Further in this manuscript we will show that without proper assessment of instrumental errors, the associated uncertainties are estimated to be over 10% for most lidar instruments presented in the study. Recent studies showed that even small deviations from ideal lidar optics can lead to large uncertainties of the retrieved depolarization products (Bravo-Aranda et al., 2016). Typically, the main source of uncertainty does not come from the detected signal noise but from





systematic errors in the optical setup of the lidar systems (Freudenthaler et al., 2009; Freudenthaler, 2016; Alvarez et al., 2006; Snels et al., 2009; Biele et al., 2000; David. G, et al., 2012). One of the most efficient method in measuring the absolute value of depolarization parameters is by implementing hardware depolarization calibration methods.

This study aims to present the available techniques developed to calibrate the lidar depolarization channels in EARLINET (David. G, et al., 2012; Freudenthaler, 2016). Moreover, methods for assessing the influence of lidar optics on depolarization

products (i.e., the assessment of the receiving optics diattenuation parameter and the rotation of the plane of polarization of the laser around the light propagation axis with respect to the PBS) in order to reduce the corresponding uncertainties (Mattis et al., 2009) are discussed.

The study also provides different experimental procedures for the assessment of several instrument parameters required to correct the lidar depolarizatioin products, stressing the effects of these parameters on the depolarization products. A first

assessment on the depolarizatoin accuracy is also provided for selected lidar instruments.

The first part of the paper describes the theoretical background, architecture, methodology and a broad description of the available calibration procedures. New techniques to retrieve the influence of different optical modules on depolarization products are presented and discussed. Techniques to assess and correct the rotation of the plane of polarization of the laser around the light propagation axis ($\alpha$) are also introduced. Section 2 describes the theoretical background based on the Mueller-Stokes

formalism used as the basis for the entire study (David. G, et al., 2013; Freudenthaler, 2016). The methodology is given in Section 3.

The second part of the paper shows results of calibrated and not-calibrated lidar depolarization profiles, several case studies from different lidar instruments in EARLINET, discussions and conclusions. Volume and particle linear depolarization ratios are presented, emphasizing the added value of calibrated depolarization channels, especially when quantitative information is

required. Section 4 shows the results and discussions and the conclusions are given in Section 5.

The practical approach of the paper is designed to present how depolarization calibration procedures are implemented. Most of the available literature is focused on the theoretical perspective of the topic and practical issues usually remain opened.

## 2 Theoretical background

The Mueller-Stokes formalism (Chipman, 2009; Ossikovski et al., 2010; Lu and Chipman, 1996) describing the lidar system

setup (shown in Fig.1) can be summarized by the following equation (Freudenthaler, 2016):

$$I_S = \eta_S \mathbf{M}_S(D_S) \mathbf{R}(y) \mathbf{M}_0(\gamma, D_0) \mathbf{F}(a) \mathbf{M}_E(\beta) I_L(\alpha, a_L) \qquad (1)$$

where bold italic fonts are used for the Stokes vectors, bold for the Mueller matrices and italic for the scalar variables. $\mathbf{I}_L(\alpha, a_L)$ is the Stokes vector of the light emitted by the laser, $\mathbf{M}_E$ is the Mueller matrix of the emission block optics, $\mathbf{F}$ represents the Mueller matrix of the atmospheric scattering volume in backscattering direction, $a$ is the polarization parameter of the

atmospheric volume described below in more detail, $\mathbf{M}_0$ is the receiving optics matrix characterized by the receiving optics diattenuation parameter $D_0$, $\mathbf{R}(y)$ is the rotation matrix, $y$ describes the optical setup type (see Fig 2.a-b). $\mathbf{M}_S$ stands for both parts of the PBS, i.e. the transmitted (subscript T) and reflected (subscript R) channels, including additional polarizing





elements after the PBS. $D_S$ is the PBS diattenuation parameter. The incident plane of the PBS is taken as the reference plane for all rotation angles around the optical axis. $\eta$ represents the calibration factor including only the electronic amplification and the optical diattenuation of the two polarizing channels and $\boldsymbol{I}_S$ is the Stokes vector for the two detected channels (i.e. $\boldsymbol{I}_R$ and $\boldsymbol{I}_T$) (see also Fig 1) where the Stokes vector describes the polarisation state of the measured channels (either transmitted or reflected). $\alpha$ is the rotation of the plane of polarization of the laser around the propagation axis (also called laser rotation),

$a_L$ is the polarization parameter of the light beam leaving the laser (the laser beam polarization purity), $\beta$ is the rotation of the emitter optics around the propagation axis, $\gamma$ is the rotation of the receiver optics around the propagation axis. In order to have a complete characterization of the lidar optics, the contribution of all latter parameters must be accounted. The technological solutions for mounting the receiving optics and the PBS are based on high precision optical mounts for all the considered lidar setups. These implementations assure high accuracy and minimization of any rotation misalignment of the optics. The

analyzed EARLINET lidars have the $\gamma$ and $\beta$ angles lower than $0.5°$ as indicated by (Bravo-Aranda et al., 2016). Since the larger uncertainties are expected for $D_O$ and $\alpha$, we neglect the effect of $\gamma$ and $\beta$ angles on these lidar systems.

    A significant simplification comes from the "v" component of the emitted Stokes vector (i,q,u,v) (Chipman, 2009; Lu and Chipman, 1996; Freudenthaler, 2016; Ossikovski et al., 2010; David. G, et al., 2012). By neglecting this component, we assume that the emitting optics does not have retardation effects. This simplification can be performed once the $\alpha$ parameter is

corrected and the $\beta$ angle is negligible. According to (Freudenthaler, 2016; David. G, et al., 2012), diattenuating and retarding optics such as dichroic mirrors should be carefully aligned as they can convert linear polarized into elliptically polarized light. Another source of elliptically polarized light could be the laser emission, but according to laser specifications, the polarization purity of commercial Nd:YAG lasers is higher than 95% and the elliptical light component of the remaining light should be even lower.

The study is focused mainly on the ***measured calibration factor*** $\eta^*$, the ***rotation of the plane of polarization of the laser around the propagation axis*** $\alpha$ and the ***diattenuation parameter of the receiver optics*** $D_0$, but also other notable parameters will be discussed (e.g. $D_S$ and $\varepsilon$). The polarizing beam splitter cross talk is usually reduced by additional polarization filters placed after the PBS on both transmitted and reflected channels. Still, the cross talk effects will be included for theoretical purposes ($D_S$). The polarization parameter of the light beam leaving the laser $a_L$ should be considered in case of instruments

where the laser polarization purity is not achieved by additional optics or where this correction is mandatory. Additional measurements are required for the assessment and correction of this parameter if this was not already provided by the laser manufacturer. Freudenthaler (2016) describes in details all terms and the variables present in Eq. (1).

    The light emitted by the laser is:

$$\mathbf{I}_L\left(\alpha, a_L\right) = \begin{pmatrix} 1 & 0 & 0 & 0 \\ 0 & c_{2\alpha} & -s_{2\alpha} & 0 \\ 0 & s_{2\alpha} & c_{2\alpha} & 0 \\ 0 & 0 & 0 & 1 \end{pmatrix} \cdot I_L \begin{pmatrix} 1 \\ a_L \\ 0 \\ 0 \end{pmatrix} = I_L \begin{pmatrix} 1 \\ c_{2\alpha} a_L \\ s_{2\alpha} a_L \\ 0 \end{pmatrix} \tag{2}$$




where

$$c_{2\alpha} = \cos(2\alpha) \text{ and } s_{2\alpha} = \sin(2\alpha) \tag{3}$$

The effects of beam expanders and steering mirrors after the laser unit can have an influence on the degree of laser polarization (David. G, et al., 2012). This optics can produce elliptical polarized light. For a more general approach we can define the emitter Stokes vector with arbitrary state of polarization that could include the effects of the emitter optics $\mathbf{I}_E$. To overcome the effects of the emitter optics, a good approach is to use direct laser emission without beam expansion and steering. For this

case, mechanical solutions to directly align the laser with respect to the receiving unit are already available. Two of the lidar instruments considered for this study send the laser radiation in the atmosphere without using any optic (MUSA and MULIS). For lidar systems that use emitter optics to send the laser radiation in the atmosphere, further investigations are needed to fully characterize the effects of $\mathbf{M}_E$ on depolarization products (Bravo-Aranda et al., 2016).

$$\mathbf{I}_E(\beta,\alpha,a_L) = \mathbf{M}_E(\beta) \cdot \mathbf{I}_L(\alpha,a_L) = T_E I_L \begin{pmatrix} i_E \\ q_E \\ u_E \\ v_E \end{pmatrix} \cdot \begin{pmatrix} 1 \\ c_{2\alpha} a_L \\ s_{2\alpha} a_L \\ 0 \end{pmatrix} \tag{4}$$

The Mueller matrix describing the atmospheric backscatter is:

$$\mathbf{F}(a) = \begin{pmatrix} F_{11} & 0 & 0 & 0 \\ 0 & F_{22} & 0 & 0 \\ 0 & 0 & -F_{22} & 0 \\ 0 & 0 & 0 & F_{44} \end{pmatrix} =$$

$$F_{11} \begin{pmatrix} 1 & 0 & 0 & 0 \\ 0 & a & 0 & 0 \\ 0 & 0 & -a & 0 \\ 0 & 0 & 0 & 1-2a \end{pmatrix} \tag{5}$$

$$a = \frac{F_{22}}{F_{11}} \tag{6}$$

Consequently, the linear depolarisation ratio of the atmospheric scattering volume ($\delta$) can be defined as

$$\delta = \frac{F_{11} - F_{22}}{F_{11} + F_{22}} = \frac{1-a}{1+a} \Rightarrow a = \frac{1-\delta}{1+\delta} \tag{7}$$



All optical elements $\mathbf{M}_O$ can be described by Mueller matrices of diattenuators $\mathbf{M}_D$ with retardation $\mathbf{M}_{ret}$ (Garcia , 2013):

$$\mathbf{M}_O\left(\gamma,D_0\right) = \mathbf{M}_D\mathbf{M}_{ret}\mathbf{M}_\gamma = T_O \begin{pmatrix} 1 & D_O & 0 & 0 \\ D_O & 1 & 0 & 0 \\ 0 & 0 & Z_O & 0 \\ 0 & 0 & 0 & Z_O \end{pmatrix} \cdot \begin{pmatrix} 1 & 0 & 0 & 0 \\ 0 & 1 & 0 & 0 \\ 0 & 0 & c_O & s_O \\ 0 & 0 & -s_O & c_O \end{pmatrix} \cdot \begin{pmatrix} 1 & 0 & 0 & 0 \\ 0 & c_{2\gamma} & -s_{2\gamma} & 0 \\ 0 & s_{2\gamma} & c_{2\gamma} & 0 \\ 0 & 0 & 0 & 1 \end{pmatrix} =$$

$$= T_O \begin{pmatrix} 1 & D_O & 0 & 0 \\ D_O & 1 & 0 & 0 \\ 0 & 0 & Z_O c_O & Z_O s_O \\ 0 & 0 & -Z_O s_O & Z_O c_O \end{pmatrix} \cdot \begin{pmatrix} 1 & 0 & 0 & 0 \\ 0 & c_{2\gamma} & -s_{2\gamma} & 0 \\ 0 & s_{2\gamma} & c_{2\gamma} & 0 \\ 0 & 0 & 0 & 1 \end{pmatrix} \tag{8}$$

with

$$T_O = \frac{T_O^p + T_O^s}{2}, D_O = \frac{T_O^p - T_O^s}{T_O^p + T_O^s}, Z_O = \frac{2\sqrt{T_O^p T_O^s}}{T_O^p + T_O^s} = \sqrt{1-D_O^2},$$
$$c_O = \cos\Delta_O, s_O = \sin\Delta_O, \Delta_O = \varphi_O^p - \varphi_O^s \tag{9}$$

where $\Delta_O$ is the retardation = differential phase shift ($\varphi$) of the p and s polarised light components and $T^p$, $T^s$ are the optics intensity transmission for parallel (p) and cross (s) linearly polarised light with respect to the plane of incidence of the PBS.

The Mueller matrix of the PBS can be defined as:

$$\mathbf{M}_S\left(D_S\right) : \mathbf{M}_R\left(D_R\right) \text{ and } \mathbf{M}_T\left(D_T\right) \text{ the reflected and transmitted components} \tag{10}$$

$$\mathbf{M}_T\left(D_T\right) = \begin{pmatrix} 1 & D_T & 0 & 0 \\ D_T & 1 & 0 & 0 \\ 0 & 0 & Z_T c_T & Z_T s_T \\ 0 & 0 & -Z_T s_T & Z_T c_T \end{pmatrix} \tag{11}$$

and an extra reflection matrix for the reflected component

$$\mathbf{M}_R\left(D_R\right) = \begin{pmatrix} 1 & D_R & 0 & 0 \\ D_R & 1 & 0 & 0 \\ 0 & 0 & -Z_R c_R & -Z_R s_R \\ 0 & 0 & Z_R s_R & -Z_R c_R \end{pmatrix} \tag{12}$$

By using a cleaned polarising beam splitter (additional polarization filters placed after the PBS to minimize the amount of residual light passing in the orthogonally polarized component - the cross talk) we can obtain

$$D_R = -1, D_T = +1 \Rightarrow D_S = \pm 1 \tag{13}$$

The rotation matrix $\mathbf{R}(y)$ can be defined as:

$$\mathbf{R}\left(y\right) = \begin{pmatrix} 1 & 0 & 0 & 0 \\ 0 & y & 0 & 0 \\ 0 & 0 & y & 0 \\ 0 & 0 & 0 & 1 \end{pmatrix} \tag{14}$$





where *y* describes the optical setup type: y= 1 for $90°$ and y= $-1$ for $0°$ (Fig 2.a-b).

With all these considerations, the detected light intensity for the p and c components Eq. (1) can be rewritten as:

$$\boldsymbol{I}_S = \eta_S \mathbf{T}_S \mathbf{T}_O \mathbf{T}_{rot} \mathbf{F}_{11} \mathbf{T}_E \boldsymbol{I}_L \left( \mathbf{G}_S + a \mathbf{H}_S \right) \tag{15}$$

with

$$\mathbf{G}_S \left( y, \gamma \right) = \left( 1 + y D_S D_O c_{2\gamma} \right) i_E - y D_S Z_O s_O s_{2\gamma} v_E \tag{16}$$

$$\mathbf{H}_S \left( y, \gamma, \beta, \alpha, \right) = D_O \left( c_{2\gamma} q_E - s_{2\gamma} u_E \right) + y D_S \left[ q_e - s_{2\gamma} \left[ W_0 \left( s_{2\gamma} q_e + c_{2\gamma} u_e \right) - 2 z_0 s_0 v_e \right] \right] \tag{17}$$

for most cases we consider $i_e = 1$, $q_e = c_{2\alpha} \, a_L$, $u_e = c_{2\alpha} \, a_L$, $v_e = 0$ and

$$W_O = 1 - Z_O c_O \tag{18}$$

## 2.1 Depolarization calibration theory

First parameter required for the calibration of the depolarization channels is the measured calibration factor $\eta^*$. This parameter includes the effects of different quantum efficiencies for the two detection modules that are part of the depolarization channels but also crosstalk of the PBS module and optics diattenuation after the calibrator. "After" refers to the light direction given by the Mueller-Stokes formalism with respect to different optical components. Different experimental methods for assessing the measured calibration factor are presented in Section 3. These methods are also used to derive other instrumental depolarization parameters like the error angle of the "$\Delta 90°$" polarizer rotation calibrator introduced later in the study ($\varepsilon$), $\alpha$ and the diattenuation parameter.

$$\eta^* = \frac{I_R}{I_T} \left( x 45° \right) \tag{19}$$

where x is a constant defined as $\pm 1$. For an ideal lidar instrument, the measured calibration factor should be equal to the real calibration factor $\eta$. For real lidar instruments, the measured calibration factor is affected by the latter instrumental depolarization parameters $(D_S, \alpha, \varepsilon, a_L, D_O,)$. To correct for these contributions, the theoretical correction factor of the measured calibration factor must be determined (K).

$$\eta = \frac{1}{K} \eta^* \tag{20}$$

The theoretical correction can be retrieved from the analytical expression by substitution of all known instrumental depolarization parameters. Part of the instrumental parameters can be determined by means of additional calibration measurements that will be detailed in the following sections.

$$K = \frac{\eta^*}{\eta} = \frac{\langle \mathbf{A}_r \left( y \right) | C \left( x 45° \right) | \boldsymbol{I}_{in} \rangle}{\langle \mathbf{A}_t \left( y \right) | C \left( x 45° \right) | \boldsymbol{I}_{in} \rangle} \tag{21}$$

For assessing K, the considered lidar setups can be described using the braket vectors. In this notation we divide the instrument optical modules in three groups: modules before the calibrator $\boldsymbol{I}_{in}$, the depolarization calibrator $C \left( x 45° \right)$ and the modules



behind the calibrator $\mathbf{A}_s(y)$. $\mathbf{A}_s(y)$ is the analyser matrix, $\boldsymbol{I}_{in}$ is the input Stokes vector that include the matrices before the calibrator and the emitted Stokes vector $\boldsymbol{I}_E$. A detailed theoretical study on different lidar setups and positions of the depolarization calibrator can be found in Freudenthaler (2016).

The measured polarization ratio $\delta^*$ can be determined by using:

$$\delta^* = \frac{1}{\eta} \cdot \frac{I_R}{I_T} \tag{22}$$

and the real polarization ratio $\delta$ can be determined using:

$$\delta = \frac{1-a}{1+a} = \frac{\delta^*(G_T + H_T) - (G_R + H_R)}{(G_R - H_R) - \delta^*(G_T - H_T)} \tag{23}$$

## 3 Methodology

### 3.1 Assessment of the measured calibration factor $\eta^*$: calibration procedures

The calibration of depolarization channels is specific to each lidar system, but the basic principles are similar for most of the instruments. The calibration of the depolarization channels consists of assessing the measured calibration factor $\eta^*$ and then applying all necessary corrections to reduce the contribution of the instrument.

In order to determine the measured calibration factor $\eta^*$, a first approach is to use the "0° calibration" or the "atmospheric calibration". Using this calibration, the contribution of the system to the final lidar depolarization products is assessed by using a low aerosol height range in the lidar signal, an altitude where only the molecular contribution could be considered. In such an atmospheric region, the total volume linear depolarisation ratio can be approximated by the well known value of the air molecule linear depolarisation ratio (Behrendt and Nakamura, 2002). Usually this calibration does not take into account all effects that have to be corrected in the depolarization profile resulting in erroneous depolarization values especially in highly depolarizing aerosol layers. Another drawback is the presence of small amounts of highly depolarizing aerosol (e.g. ice crystals) in the assumed clean range that can easily lead to large errors in the depolarization products (Freudenthaler et al., 2009; Freudenthaler, 2016). Other calibration techniques include the use of depolarization optics in the receiver to calibrate the lidar gain ratio (Winker et al., 2007) or the use of three lidar signals (cross, parallel and total) to calibrate the depolarization products. This method makes use of two altitude ranges – high depolarization and low depolarization load - to extract the calibration constant for the calibration channels (Reichardt et al., 2003).

A reliable solution to calibrate the depolarisation measurements is represented by the "45° calibration". This calibration implements a 45° rotation of the depolarization analyzer (PBS and the PMTs) with respect to the polarization plane of the laser in order to equalize the light intensity in the cross and parallel channels. When comparing the calibration signals, the ratio between the transmitted and reflected signals reflects the contribution of optics and electronics in the lidar receiving unit. The implementation of these methods will be further described in this study.

The main source of uncertainty involved in this kind of calibration is represented by the accuracy in determining the rotation of 45° with respect to the true zero position of the PBS. The less is this accuracy the large is the errors in estimating the calibration constant. A better solution is to use two subsequent measurements performed by rotating the depolarization analyzer





at $\pm 45°$ with respect to the default measuring position (David. G, et al., 2012). This calibration is called the "$\pm 45°$ calibration". The calibration constant is determined by using the geometric mean of the two $\pm 45°$ measurements. The two measurements are designed to compensate each other even for cases where the $45°$ rotation uncertainty is large with respect to the initial zero position given by the PBS (Freudenthaler et al., 2009). Since for the "$\pm 45°$ calibration", the initial zero position reference

is not important, a more general solution is to use two subsequent measurements performed by rotating the depolarization analyzer with an exact $90°$ difference between each other. This calibration method is called the "$\Delta 90°$ calibration" and the output is similar with the one from the $\pm 45°$ calibration. The "$\pm 45°$ calibration" can be considered a particular case of the "$\Delta 90°$ rotation calibration" since the only constrain of this calibration is the $90°$ angle between the two measurements.

Technically, the "$\Delta 90°$ calibration" can be implemented by using a mechanical rotator (holder), that rotates the optical

components at fixed $\Delta 90°$ angles. This calibrator will be further called the "$\Delta 90°$ mechanical rotation calibrator". A similar approach (same output) can be considered if we use a HWP for accurately rotating the emitted or collected light at $\Delta 90°$. The advantage is that while the mechanical rotator can only be placed in the reception unit (in front of the receiving optics or in front of the PBS), the HWP module can be also placed at the emission, in front and after the emission optics. This calibrator will be further called the "$\Delta 90°$ HWP calibrator". A third approach of the "$\Delta 90°$ calibration" is the use of an additional linear

polarizer that can be rotated at fixed $\Delta 90°$ angles. In this case, the $\Delta 90°$ rotation will be replaced by the additional linear polarizer. According to its position in the optical chain (in front of the telescope, receiving optics or the PBS) the calibration can account for all lidar optics placed after the polarizer (e.g. receiving optics, PBS, PMT). This is also valid for the other calibrators. Further on, this calibrator will be called the "$\Delta 90°$ polarizer rotation calibrator". In order to perform the latter calibration, the 'zero' position of the optical module in respect to the relative position of the PBS must be determined and

corrected for. For this, the $\Delta 90°$ rotation calibration requires an extra measurement set to assess the error angle caused by the offset between the calibrator and the zero position of the PBS. Table 1 summarizes main advantages and disadvantages when using different calibration techniques for the $\Delta 90°$ calibration.

E.g. the error angle of the calibration setup ($\varepsilon$) must be estimated to allow a reliable measurement of the calibration constant when using the $\Delta 90°$ polarizer calibration. The calibration error angle $\varepsilon$ has to be corrected either mechanically before the

measurements or analytically after the measurements.

In order to determine $\varepsilon$, a set of two relative $\Delta 90°$ measurements is required. The polarizer is placed in a random position relative to the polarization plane of the receiving optics. Two measurements will be performed with the polarizer rotated precisely at $\pm 45°$ from the $\varepsilon$ angle.

By simple mathematical considerations, Freudenthaler (2016) shows that the error angle can be determined with a good

approximation from:

$$Y\left(\varepsilon, K\right) = \frac{\eta^*_{pol}\left(y, +45°, \varepsilon, K\right) - \eta^*_{pol}\left(y, -45°, \varepsilon, K\right)}{\eta^*_{pol}\left(y, +45°, \varepsilon, K\right) + \eta^*_{pol}\left(y, -45°, \varepsilon, K\right)} \tag{24}$$

and

$$\varepsilon = \frac{1}{2}\arcsin\left[\frac{1}{K}\tan\left(\frac{\arcsin\left(Y\left(\varepsilon, K\right)\right)}{2}\right)\right] \tag{25}$$





Note that the assessment of the calibrator rotation angle can only be performed in stable atmospheric conditions.

Another method to correct for the error angle is by looking at relatively clean and stable atmosphere regions and minimize the cross polarized signal. In addition, one would look at minimizing the difference at complementary angles ($\pm$) from the assumed angle and iterating (this assumes that there is no ellipticity in the laser beam or retardation effect in the receiver).

The particularity of the $\Delta 90°$ calibration also enables the assessment of other instrumental depolarization parameters required for the theoretical correction of the calibration factor - Eq. (21). Eq (24) and Eq (25) show how the $\Delta 90°$ calibration is used to assess the error angle ($\varepsilon$) of the $\Delta 90°$ polarizer rotation calibrator. Subsection 3.3 shows how the $\Delta 90°$ calibration is used to assess the diattenuation parameter for individual optical modules. Subsection 3.4 shows how this calibration is used to assess the rotation of the plane of polarisation of the laser around the propagation axis ($\alpha$).

This study will present the implementation of all these calibration methods, according to specific lidar setups in the EAR-LINET network but also the methods to assess different instrumental depolarization parameters required to correct the measured calibration factor. A comparison between these different calibration methods, advantages and disadvantages, and possible error sources is also discussed and analyzed.

### 3.2    Assessment of the measured calibration factor $\eta^*$: experimental solutions

**3.2.1    $\Delta 90°$ mechanical rotation calibrator and HWP calibrator**

The first experimental setup for the lidar depolarization calibration is based on the calibrator module placed in front of the polarizing splitter (C1 in Fig 3.a) or in front of the receiving optics (C2 or C2' in Fig 3.a). The calibrator consists of a high precision mechanical rotator implementing rigid rotations of the PBS and PMTs at + 45° and - 45° with respect to the default measuring angle (considered the 0° position) (Fig 3.c). By rotating the calibration module at + 45° or - 45°, the light

intensities in the transmitted and reflected paths are equalized independent of the atmospheric depolarization. The "$\Delta 90°$ calibration" provides the measured calibration factor $\eta^*(\pm 45°)$. E.g., the Athens EARLINET station (Kokkalis et al., 2013) operates depolarization lidars using a mechanical rotator in front of the PBS for the "$\Delta 90°$ calibration" (Mamouri et al., 2012). Another approach with similar results as the $\Delta 90°$ mechanical rotation calibrator is the use of a HWP to rotate the plane of polarization of the collected light to the desired angles (in this case $\pm 45°$). This calibrator is called the $\Delta 90°$ HWP calibrator.

The HWP rotator calibrator has the same effect and uses the same formulas as the mechanical rotator calibrator. In addition to the mechanical rotation calibrator, the HWP calibrator can also be placed in the emission block of the lidar system (C3 and C4 in Fig 3.a). This calibration method is used e.g. by the Munich (MULIS) (Freudenthaler et al., 2009) and Potenza (MUSA) lidar systems (Madonna et al., 2011). In both these systems the calibration module consists of a HWP rotator placed in front of the PBS, which rotates the plane of polarization of the light by $\pm 45°$ with respect to the default polarization angle. The same

type of calibrator can be also implemented by using a stepping motor rotation mount or a HWP mount which is placed in a holder with fixed and accurate positions at 0° and $\pm 45°$ (or multiple positions) (Fig 3.b).

An advantage when using this method is that measurements are not affected by the calibrator itself. The $\Delta 90°$ mechanical rotation calibration introduces an angle error ($\Psi$), always present in the measurement, whereas for cases where the $\Delta 90°$ HWP





calibrator is removed after the calibration procedure, any errors introduced by the multi-angle polarizer calibrator or the HWP calibrator will not influence the measurements.

### 3.2.2 $\Delta 90°$ polarizer rotator calibrator

The third approach for the lidar depolarization calibration at $\Delta 90°$ is the use of a linear polarizer. This type of calibrator

can be implemented by using the mechanical rotating ring or the stepping motor rotation mount used for the optical rotator calibrator (Fig 3.b). The calibrator can be placed in front of the polarizing splitter ($C_1$ in Fig 3.a) in front of the receiving optics ($C_2$ or $C'_2$ in Fig 3.a). Several EARLINET lidar systems are using this calibration technique, in different versions. The cost-efficiency and simple design of this calibrator makes it easy to implement and also easy to use. Moreover, as it is a quite compact optical element, typically, it does not take much space to fit in the majority of the lidar optical chains. E.g., Leipzig and

Evora lidar stations operate POLLY-XT multi-wavelength depolarization Raman lidar systems - (Althausen, 2013) - with cross and total depolarization channels at 532 and 355 nm (Leipzig) and 532nm (Evora) (Fig 2.c). Calibration of the depolarization channels for these instruments is performed using the $\pm 45°$ rotatable polarizer, placed in front of the detection optics or near the telescope's field stop. For this case, the acceptance angles of the polarizer used for calibration must be accounted for since the calibration requires high extinction ratios.

All experimental setups presented in this section are based on the "$\Delta 90°$ calibration" procedure, therefore the methodology describing the assessment of the lidar depolarization calibration constant is similar for all calibrators described in the study. For all these calibration procedures, the measured calibration factor can be derived from the geometric mean of the two consecutive measurements at -45° and +45°.

For a general approach, the theoretical framework describing the assessment of the measured calibration factor for the lidar

depolarization channels is described in details by Freudenthaler (2016).

### 3.3   Assessment of the diattenuation parameter $D_O$

This section provides information on how the $\Delta 90°$ calibration can be used to assess the diattenuation parameter of the receiving optics. This measurement is important for lidar instruments that use depolarization calibration techniques in front of the PBS. For this case, one additional measurement is required to assess the contribution of the receiving optics ($D_O$) to the

depolarization products each time changes are performed to the receiving optics or the laser. The parameter will be later used to correct the measured calibration factor and to assess the $H_S$ and $G_S$ parameters.

By comparing the calibration values obtained using the two calibrators placed in front and after a specific optical module, the investigator can assess its depolarization effects (the diattenuation parameter (Mattis et al., 2009)). Simulations performed by (Bravo-Aranda et al., 2016) shows that the effects of the diattenuation on depolarization products are highly significant. By

using this method, we can correct the diattenuation effects for either the receiving optics (if the calibration modules are placed in front and after the receiving optics), the emitting optics (if the calibration modules are placed in front of the emitting optics and in front of the receiving optics) or both. Once the diattenuation parameter is known, we can correct for its effect regardless of the calibrator's default position in the optical chain.



Several systems, such as lidars operated by Munich (MAISACH) (Freudenthaler et al., 2009), Granada (MULHACEN) (Guzman et al., 2013) or Bucharest stations (RALI) (Nemuc et al., 2013), have two or more depolarization calibration methods implemented. Both Granada and Bucharest stations run multi-wavelength Raman depolarization lidars, measuring two depolarization channels at 532nm and a 90° setup (Fig 2.a). The depolarization calibration setup consists of a set of two calibration

modules/techniques, designed to evaluate the contribution of certain lidar sections on the output depolarization products ($D_O$).

The first calibrator is a mechanical rotator placed in front of the PBS - $C_1$ in Fig 3.a (for the $\Delta 90°$ calibration) with a rotation accuracy better than $\pm 0.1°$ - $\eta^*_{\text{after}}$. The second calibrator consists of a linear polarizer mounted in front of the telescope's field-stop or in front of the receiving block ($\mathbf{M}_O$) of the lidar system ($C_2$ and $C'_2$ in Fig 3.a) - $\eta^*_{\text{before}}$. A mechanical mount allows the polarizer to rotate by a fixed $22.5° \pm 0.05°$ rotating steps (Fig 3.b). By comparing the results obtained using the two calibrators,

the diattenuation of the optical elements in between ($\mathbf{M}_O$) can be determined. For lidar systems like the ones of Potenza and Munich, the diattenuation effect of the receiving optics ($\mathbf{M}_O$) is known to be low due to a particular design of the optical module (optimized angles) and special manufactured optical components designed to reduce the diattenuation effects. In the case of Bucharest and Granada lidar systems, the influence of the receiving optics is known to have a greater impact on the depolarization products - see Table 2. The diattenuation effects can be corrected in the post measurement analysis, if the $D_O$

parameter of the considered optical module is determined. The diattenuation of the receiving optics can be easily determined by assessing the ratio:

$$\eta^*(D_0) = \frac{\eta^*_{\text{before}}(\pm 45°)}{\eta^*_{\text{after}}(\pm 45°)} = \frac{1 + D_0}{1 - D_0} \tag{26}$$

leading to

$$D_O = \frac{\eta^*_{\text{before}}(\pm 45°)/\eta^*_{\text{after}}(\pm 45°) - 1}{\eta^*_{\text{before}}(\pm 45°)/\eta^*_{\text{after}}(\pm 45°) + 1} \tag{27}$$

**3.4   Assessment of and correction for the laser rotation $\alpha$**

Orientation of the plane of polarization of the laser around the propagation axis is not accurately provided by laser manufacturers. The mechanical assembly between the laser and the receiver optics can often contribute to the rotation between the laser emission and the PBS since the accuracy of these assemblies is lower that the alignment accuracy of the optical elements. The alignment mechanism of the lidar instrument used to tilt the laser beam could also be a source of variability and uncertainty.

Considering these limitations, it is important to assess the laser rotation around the propagation axes ($\alpha$). For large values, the effects of this parameter are significant and must be accounted for. To assess the effects of $\alpha$ on the polarization ratio and to determine the best experimental solution to correct for this parameter, several simulations were performed for the Bucharest lidar system. The main goal of these simulations is to stress the effects on the rotated input Stokes vector transmitted on different optics and to show the reader how different correction methods are affecting the real polarization ratio. Fig 4.a-b shows

simulations of measured polarization ratio as a function of $\alpha$. The simulations demonstrate that for angles smaller than 3°, the effect on the calibrated signal ratio is negligible (Fig 4.b). The effects induced on the measured polarization ratio increase dramatically for higher values of the angles (Fig 4.a). According to these simulations, the effects of $\alpha$ are also dependent on





atmospheric depolarization: as atmospheric depolarization decreases, the dependence between the retrieved measured polarization ratio and $\alpha$ is more significant. In real situations, the optical misalignment for $\alpha$ will not exceed 10-15°, but in order to present a complete dependency of the calibrated signal ratio, Fig 4.a shows the behaviour of the latter for $\alpha$ ranging from 0° to 180°. Simulated measured calibration factor ($\eta$*), obtained using the mechanical rotation in front of the PBS (Fig 4.c) show the dependency of the latter with atmospheric depolarization for $\alpha$ ranging from 0° to 10°. This dependency alters the experimental retrieval of the measured calibration factor whenever $\alpha$ is considerable large ($\alpha>5°$). A good practice would be to assess and correct for the $\alpha$ angle before performing the depolarization calibration.

### 3.4.1 Assessment of $\alpha$ parameter

In order to determine $\alpha$, we apply the same principles as for assessing the calibrator rotation angle - $\varepsilon$ for the $\Delta 90°$ polarizer rotator calibrator (Alvarez et al., 2006; David. G, et al., 2012; Freudenthaler, 2016).

Simulations in Fig 5.a-b show a strong dependency between Y and two parameters: the polarization parameter of the atmospheric volume and the diattenuation parameter. The dependency of Y with the polarization parameter of the atmosphere does not affect the assessment of $\alpha$ drastically, since all the retrievals are performed in an aerosol free height, where the atmospheric depolarization is minimal. Although simulations reveal a notable link between Y and the latter parameters, this dependency becomes negligible as the $\alpha$ value decreases. This particularity allows a highly accurate experimental correction of $\alpha$ by applying an iterative procedure: after the first iteration (first $\alpha$ assessment and experimental correction), the effects of the polarization parameter (atmospheric depolarization) and the diattenuation on the second $\alpha$ assessment are decreased and the correction becomes more and more accurate. The second iteration is performed for smaller $\alpha$ values, therefore having a better accuracy. After several iterations, the retrieved $\alpha$ value will be close to zero. The iteration method does not apply to the analytical correction of $\alpha$.

### 3.4.2 Correction for the $\alpha$ parameter

The analytical correction of $\alpha$ can be performed by using Eq. (16), (17) and (21). According to simulations (Fig 5.c), uncertainties for the analytical correction of $\alpha$ can go up to 25% for a 10° initial offset and a diattenuation value $D_O = 0.25$. The experimental correction of $\alpha(Y)$ can be performed either by rotating the PBS in the WSU (without or together with the receiving optics) or by rotating the plane of polarization of the collected light using a HWP placed in front of the PBS or in front of the receiving optics (in the case of one wavelength lidar instruments or systems with separate optics for the depolarization channels).

**In front of the PBS:**

For lidar systems designed to use the mechanical rotation calibrator in front of the PBS, the most efficient technique to correct for $\alpha$ is by rotating the PBS according to its determined value. Equivalent results can be obtained also by rotating the laser polarization plane by means of a HWP module placed in front of the PBS. For the $\alpha$ correction, the compensation angle of the correction module will be considered $\varepsilon$'. Simulations of the calibrated signal ratio corrected with $\varepsilon$' are presented in Fig 6.a (for $\alpha$=-$\varepsilon$'). The results show that for $\varepsilon$'=10°, the correction error is reaching 3% from the absolute value of the calibrated





signal ratio. This error is most probably caused by the method itself: when using the mechanical rotation or HWP rotation in front of the PBS, we compensate for $\alpha$ after the light has gone through the receiving optics ($\mathbf{M}_O$). The effects introduced by $\alpha$ in the receiving optics, as the collected backscattered light is guided toward the PBS, are not removed by this correction. We must stress that in order to perform a comprehensive simulation, the diattenuation parameter of the receiving optics was

considered 0.23 (measured diattenuation for the Bucharest RALI lidar system - Table (2)) and the atmospheric depolarization 0.05 (since higher values for the atmospheric depolarization will not drastically alter the correction error).

**In front of the receiving optics:**

For lidar systems designed to use a mechanical rotation calibrator in front of the receiving optics, the optimal technique designed to correct for $\alpha$ is by rotating the receiving optics accordingly. This technique is considered to be better since by

rotating both the receiving optics and the PBS, all the effects introduced by $\alpha$ in the receiving optics are compensated.

For the case of one wavelength emission, a HWP calibrator placed in front of the receiving optics can also be used to correct for $\alpha$. In case of lidar instruments having different emission axes at different wavelengths, the correction could be performed at the emission. For this case, the simulations show that the linear depolarization ratio error is less than 0.1% for a 10° offset - (Fig 6.b).

**4   Results and Discussions**

Numerous optical components inside the lidar's emission and receiving units can lead to large systematic errors of the atmospheric depolarization values (Bravo-Aranda et al., 2016). Methods designed to assess and correct instrumental effects on the depolarization channels are constantly under development. The volume linear depolarization ratio profiles show significant improvements when reliable and accurate depolarization calibration techniques are used. The impact of the calibration is mostly

visible in the low aerosol height ranges, where the rather low molecular contribution is usually added to the systematic error of the instruments. The particle linear depolarization ratio profiles are also improved by the calibration, although in this case, the uncertainties also include the contribution of the aerosol backscatter coefficient (Freudenthaler et al., 2009).

**4.1   The measured calibration factor $\eta$\***

The measured calibration factor and diattenuation values for several calibration methods are presented in Table 2. $\eta$\*$_{before}$

represents the measured calibration factor value retrieved using the $\Delta 90°$ polarizer rotator calibrator placed in front of the receiving optics and $\eta$\*$_{after}$ represents the measured calibration factor value retrieved using either the $\Delta 90°$ HWP calibrator or the $\Delta 90°$ mechanical rotation calibrator, placed in front of the PBS. Deviation values are retrieved either from consecutive measurements collected during a limited time interval, either from the variability of the measured calibration factor in the selected altitude interval.





### 4.2 The diattenuation parameter $D_0$

The diattenuation parameters for the Potenza and Munich systems show values one order of magnitude lower than for the rest of the lidar systems (Table 2). These values are the result of a special optical design of the WSU (Freudenthaler et al., 2009), combined with custom made optics specially designed to have low diattenuation values. Fig 7.a-b shows the measured

calibration factor retrieved using two experimental techniques for the Granada and Bucharest systems in order to extract the diattenuation value as presented in Section 3.3. The difference between the $\eta^*_{before}$ and $\eta^*_{after}$ represents the effect of the receiving optics diattenuation parameter ($D_0$) on the depolarization value. Fig 7.a-b illustrates the height dependence of the measured calibration factor, retrieved by using two depolarization calibration modules. The results show that for the presented altitudes, the height dependence of the calibration value is not significant (for neither the Bucharest nor Granada lidar systems).

For the Bucharest system, the measured calibration factor profiles from 1 to 3.5km show higher stability for the mechanical rotator retrievals ($\eta^*_{after}$) in respect to the polarizer rotation retrievals ($\eta^*_{before}$). This stability difference could be caused by the presence of atmospheric layers with higher depolarization signature in the investigated range, changing rapidly with time. For the Granada system, the profiles show the same stability for both $\eta^*_{before}$ and $\eta^*_{after}$. The height dependence for the measured calibration factor could be used as a good indicator of potential problems in the optical layout of the lidar system.

### 4.3 Rotation of the plane of polarisation of the laser ($\alpha$)

One of the parameters having a significant impact on depolarization products is the rotation of the plane of polarisation of the laser with respect to the PBS: $\alpha(Y)$. According to numerical simulations already presented, correction of the $\alpha$ parameter can be achieved by using a mechanical rotator or a HWP placed in front of the PBS or in front of the receiving optics. A third option is the post measurement analytical correction performed once the $\alpha$ parameter is retrieved. Further on, in order to

perform comparisons between corrected and not corrected lidar profiles, we will only consider the post measurement analytical correction for $\alpha$.

    Fig 7.c shows results of the assessment and correction of the $\alpha$ parameter for the Bucharest system, using the mechanical rotator in front of the PBS. The correction is performed by iterative steps by means of rotating the linear analyzer (PBS) in accordance with measured $\alpha$ values. The values show that measured $\alpha$ reaches a value equal to -0.04° after four iterations.

The impact of the $\alpha$ correction on the depolarization profiles can be easily emphasized for the post measurement analytical correction, presented in section 3.4.2. Fig 8.a-c shows an example of volume and particle linear depolarization ratios from $26^{th}$ of September 2013, measured by the Bucharest lidar system RALI (Nemuc et al., 2013). The range corrected time series for 532nm total (parallel + cross) show stable layers in the lower troposphere and ice clouds above 8km - Fig 8.a - red vertical lines show the averaged time period considered for the calculation of the volume linear depolarization profiles. The non-calibrated

volume linear depolarization profiles (the ratio of the two signals) show values reaching up to 0.27 in the ice cloud and 0.12 in the free troposphere (Fig 8.b). The calibrated profile ($\eta^*$, $a_L$, $D_O$ corrected, no $\alpha$ correction) shows lower values in the free troposphere and values reaching 0.42 in the ice cloud. After correction of $\alpha$, the volume linear depolarization values reaches out to 0.40 in the cloud and close to the molecular in the free troposphere (Sassen et al., 2007), (Sassen, 2005) (for this





case $\alpha = 10°$). Table 3 shows non-calibrated and calibrated (including $\alpha$ correction) volume linear depolarization retrievals in two cases: in the cloud layer and in the free troposphere. For the free troposphere, the initial values are over 10 times larger than for the calibrated profiles. Values for the calibrated profiles with and without alpha correction show a small difference in the cloud layer (0.02), but larger differences are observed in the free troposphere, where the volume linear depolarization

is of the same order of magnitude with the $\alpha(Y)$ corrections. The systematic errors associated to the not-corrected volume depolarization profile are larger in comparison with errors associated to the calibrated, $\alpha$ corrected profile. For not-corrected profiles, errors associated to each instrumental depolarization parameter must be assumed to be large (Bravo-Aranda et al., 2016). E.g. for unknown $\alpha$ values, the associated systematic error should be in the order of $\pm 10°$. For known $\alpha$ values the associated systematic error was determined to be $\pm 1°$. More details on error assessment can be found in (Freudenthaler et al.,

2009). According to Fig 8.b, the associated error bars show a significant improvement once the calibration and corrections are performed: from 0.1 for the "not corrected data" profile, to 0.06 for the "no $\alpha$ correction" profile, to 0.01 for the "corrected data" profile (for altitudes reaching 8km). A detailed description on the assessment of lidar depolarization uncertainty can be found in (Bravo-Aranda et al., 2016).

### 4.4    Selected cases of calibrated profiles in the EARLINET framework

In order to make a first estimate on the depolarization accuracy of the discussed lidar instruments and to emphasize the importance of the depolarization calibration for the long range transport and aerosol typing studies, several experimental results obtained using calibrated depolarization lidar instruments from different EARLINET stations are presented and discussed.

The data shows only the corrected depolarization profiles since many lidar systems provide automatic or hardware corrected depolarization products. Still, the calibrated depolarization products selected for this section use same calibration techniques

as presented in the current study. A comparison between corrected and uncorrected profiles is not required since the purpose of this section is to give an estimate on the accuracy of depolarization products and to present the importance of calibrated depolarization lidar products in long range transport studies. The measurements are performed on an extended time scale, so that statistical noise becomes negligible (vertical red delimiters over the range corrected signals - RCS - mark the averaged periods).

Measurements performed using the Granada lidar system (Mulhacen) in July 2012 show the presence of a distinct layer between 2.5 and 5km (Fig 9.a-c). The particle linear depolarization ratio shows high values in the aerosol layer (0.223) and levels close to the molecular depolarization in the low aerosol height ranges. The back trajectories model indicates that the corresponding air mass originates in Northern Sahara, and was transported for several days over NW Africa and the Atlantic Ocean (Fig 9.d). According to the back trajectories and the particle linear depolarization values retrieved for these altitudes,

the aerosol present in the air mass is probably mineral dust.

The RCS from the Potenza lidar system (MUSA) for August 2012 presents a strong aerosol intrusion above the PBL (Fig 10.a). The non-homogenous layer between 2 and 5km has a volume linear depolarization ratio reaching 0.18 and the particle linear depolarization ratio around 0.31 (Fig 10.b-c), indicating a case of mineral dust. The back-trajectory model shows that the air-masses originate from the Sahara regions, being transported for several days over the desert and the Atlantic Ocean (Fig





10.d). A detailed analysis including the particle linear depolarization ratio and back trajectories can also provide information regarding not only the nature of the aerosol layers, but also about the age and purity of aerosol particles (Tesche et al., 2011; Gross et al., 2011; Muller et al., 2003).

The case selected for the Munich lidar system (Maisach) refers to the Eyjafjallajökul volcanic eruption occurred during April 2010. The range corrected time series (Fig.11.a) highlight a distinct layer ranging from 2.3 up to 2.8km. The one hour mean value of the particle (volume) linear depolarization ratio measured in this layer is 0.38 (0.34). The values are consistent with typical values retrieved for fresh volcanic ash (Hervo et al., 2012; Hogan et al., 2012; Pappalardo et al., 2012). The presence of fresh volcanic ash is also confirmed by the back-trajectories: air masses originating from Southern Iceland, very close to the Eyjafjallajökul Volcano, detected over Munich 48 hour after the eruption (Fig 11.d).

The cases presented above emphasize the importance of calibrated depolarization lidar products in aerosol typing. All cases were selected in order to highlight different atmospheric layers and environmental conditions (mineral dust, volcanic ash, ice crystals). The associated errors were determined by each EARLINET station, according to their own internal error assessment procedures. For most cases, uncertainties related to the systematic errors for the calibrated volume linear depolarization profiles are within 0.01 - 0.02 for all heights up to 8-9km. The larger uncertainties of the particle linear depolarization profiles with respect to the volume are mainly caused by the backscatter profile, which is required to perform the retrievals. The uncertainties related to the backscatter profiles are the result of additional assumptions required to perform the inversions (especially during daytime) - lidar ratio profile and calibration values (Nemuc et al., 2013; Kovalev and Eichinger, 2004). Although the statistical error is negligible (averaged profiles), statistical and systematic depolarization errors are included for some cases.

## 5 Conclusions

This paper presents an extended analysis of various depolarization calibration techniques, specific to depolarization lidar systems within the EARLINET network. The calibration modules were analyzed in respect to two criteria: **the type of the calibrator** and **it's placement inside the optical chain**. Different schemes for assessing and correcting the rotation of the plane of polarization of the laser ($\alpha$) are presented. A method to retrieve the effective diattenuation of the receiving optics is discussed and analyzed as well.

The two described calibration methods (calibrator in front of the PBS and the calibrator in front of the receiving optics) proved reliable as technical solutions for the "$\Delta 90°$ calibration". The advantage when using the calibrator in front of the receiving optics is that depolarization products are also corrected for the influence of the receiving optics, while the methods that use the calibrator in front of the PBS allows to take into account only the influence of lidar modules after the calibrator, throughout the optical path. From experimental point of view, the primary design includes a mechanical rotator for the PBS, a HWP used to rotate the plane of polarization of the collected light and an extra polarizer, used to rotate according to the requirements of the calibration measurements. All calibration designs proved to be effective and the results showed significant improvements after the calibration procedures were applied. Among the several calibration schemes described in this study, the $\Delta 90°$ HWP calibrator in front of the receiving optics proved to be the most reliable. The advantages of this type of calibrator




can be summarized as follows: effectiveness when implementing the calibrator, cost efficient - the extra modules needed to mount the calibrator are cheap and easy to use. The output of the calibrator includes the contribution of the receiving optics, and errors related to the calibrator itself do not influence the measurements since the modules will be removed after performing the calibration.

Second part of the study was related to the impact of the rotation of the plane of polarization of the laser around the propagation axis with respect to the PBS ($\alpha$). The effects of $\alpha$ on the final depolarization products and the efficiency of two correction schemes were discussed and analyzed: correction in front of the PBS or in front of the receiver optics by means of a HWP or a mechanical rotator. The efficiency of the second procedure is significantly better, and the errors associated with the correction procedures are much lower than the ones obtained by rotating only the PBS. The drawback of the HWP in front of

the receiving optics is related to the number of lidar systems it can be applied to.

The combination of multiple calibration methods proved to be important in assessing the diattenuation parameter of the receiving optics. This measurement is important for lidar instruments that use depolarization calibration techniques in front of the PBS. In this setup, the diattenuation of the receiving optics is not taken into account by the measured calibration factor, resulting in erroneous depolarization profiles and high systematic errors. The assessment of the diattenuation parameter is

designed to compensate for this drawback. Once the diattenuation parameter is known, the user can correct for its effect regardless of the calibrator's default position in the optical chain.

The improvements in the depolarization values retrieved for the aerosol layers and ice clouds (where the particle depolarization reaches typical values up to 0.35-0.45±0.02) as well as in the free troposphere (where the volume linear depolarization shows values around 0.01±0.01) are visible for calibrated ($\eta^*$) and corrected ($\alpha$, $D_O$, $a_L$, $D_S$) depolarization profiles - Section

4.3. These values indicate that for calibrated + corrected signals, the depolarization accuracy at 532nm is better than ±0.015. The study also shows how the associated systematic errors are reduced by one order of magnitude when proper corrections are applied to the polarization profiles.

Presented case studies show calibrated + corrected depolarization lidar products for selected lidar stations. The calibrated depolarization profiles at 532 nm show values that fall within a range of values that are generally accepted in the literature. In

the low aerosol height ranges, where the impact of the calibration procedures is more obvious, the volume linear depolarization ratio shows values close to the molecular level: $\delta = 0.01 \div 0.03 \pm 0.015$ for all lidar instruments(Behrendt and Nakamura, 2002). Considering that for most cases presented in the study, the low aerosol height ranges are not aerosol free - small amounts of highly depolarizing aerosol could affect the profiles (e.g. ice particles) - it is safe to conclude that based on the low aerosol height range values, the depolarization accuracy estimate at 532nm is better than ± 0.03 for all presented case studies. This is

only an estimate since for a complete assessment of the lidar accuracy, extended studies are required for each lidar instrument (Freudenthaler, et al., 2016b).

This study emphasizes the need of implementing calibration and correction procedures for the retrieval of depolarization products, homogeneously for the entire EARLINET network. On the other side it is also fundamental to adapt the selected procedures to the different types of lidar systems operating within EALRINET.





## Appendix A: Acronyms and shortcuts

| | |
|---|---|
| $a$ | polarization parameter of the atmospheric volume |
| $a_L$ | polarization parameter of the light beam leaving the laser |
| $\alpha$ | rotation of the plane of polarization of the laser around the propagation axis (laser rotation) |
| $\beta$ | rotation of the emitter optics around the propagation axis |
| $\gamma$ | rotation of the receiver optics around the propagation axis |
| $c_\varepsilon$ | $\cos(\varepsilon)$ |
| $s_\varepsilon$ | $\sin(\varepsilon)$ |
| $\varepsilon$ | error angle of the $\Delta 90°$ calibration setup |
| $\psi$ | rotation of the calibrator around the light propagation axis |
| $\delta$ | linear depolarization ration of the atmospheric scattering volume, volume linear depolarization ratio (LDR), real polarization ratio |
| $\delta^*$ | calibrated signal ratio including cross talk and alignment errors, measured polarization ratio |
| $\delta^p$ | particle linear depolarization ratio (PDR) |
| D | diattenuation parameter |
| $\eta_{T,R}$ | electronic amplification of individual transmitted/reflected channels |
| $\eta$ | calibration factor including only the electronic amplification and the optical diattenuation of the polarizing beam splitter, real calibration factor |
| $\eta^*$ | measured calibration factor of the polarization channels, the calibration factor including the cross talk from optics before the polarizing beam splitter and from system alignment errors |
| $\mathbf{M}_s$ | Muller matrix of the polarizing beam splitter |
| $\mathbf{M}_{T,R}$ | Muller matrix in the transmission and reflection path |
| $T_S$ | transmission of matrix $\mathbf{M}_S$ for unpolarised light |
| $T^{p,s}$ | |
| $R^{p,s}$ | intensity transmission and reflection coefficients of the polarizing beam splitter for parallel and perpendicular linearly polarised light with respect to the plane of incidence |



| | |
|---|---|
| **F** | Muller matrix of the atmospheric scattering volume in backscattering direction |
| y | optical setup type for the cross and parallel lidar configuration. For y=-1 we have the 90° setup and for y=1 we have the 0° setup (see Figure.2). |
| $\Delta$ | differential phase shift of the p and s polarised light |
| $\phi^{p,s}$ | phase of the p and s polarised light |

*Acknowledgement.* This work was supported by the European Community's FP7-INFRASTRUCTURES-2010-1 under grant no. 262254 – ACTRIS and by a grant of the STAR–ESA Programme 55/2013-CARESSE, Grant from Portuguese Science Foundation (FCT) SFRH/BPD/81132/2011.



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





**Table 1.** Δ90° calibration methods. '+' shows advantage and '-' shows disadvantage over other presented methods

| Type | SNR | effect on measurements | position | automated |
|------|-----|------------------------|----------|-----------|
| Δ90° mechanical rotation calibrator | + | not removed after calibration | in receiving unit | + |
| Δ90° HWP calibrator | + | removed after calibration | also in emitter unit | + |
| Δ90° polarizer rotator calibrator | - | removed after calibration | in receiving unit | + |

**Table 2.** Calibration values for EARLINET lidar systems performing depolarization calibration measurements

| Site | $\eta_{\text{after}}$ | $\Delta\,\eta_{\text{after}}$ | $\eta_{\text{before}}$ | $\Delta\,\eta_{\text{before}}$ | $D_O$ | err$\Delta\,D_O$ |
|------|-----------------------|-------------------------------|------------------------|--------------------------------|-------|------------------|
| Granada | 0.14 | ±0.03 | 0.24 | ±0.03 | 0.35 | ±0.04 |
| Bucharest | 1.15 | ±0.08 | 1.9 | ±0.1 | 0.227 | ±0.1 |
| Potenza | 22.67 | ±0.10 | 25.3 | ±0.10 | 0.055 | ±0.01 |
| Athens | 0.054 | ±0.01 | - | - | - | - |
| Leipzig | - | - | 0.089 | ±0.01 | - | - |
| Munich | 42.2 | ± 0.4 | 47.5 | ± 0.9 | 0.059 | ± 0.015 |

**Table 3.** Volume linear depolarization values of calibrated and non-calibrated retrievals for the RALI lidar system on $26^{th}$ of September 2013.

| | Not-calibrated prfiles | $\eta_{pol}^{*}, a_L, D_O$ corrected, no $\alpha$ correction | $\eta_{pol}^{*}, a_L, D_O$ corrected, $\alpha$ correction | Difference: non calibrated and/ calibrated profiles ($\alpha$ corrected) |
|------|------------------------|-------------------------------------------------------------|----------------------------------------------------------|--------------------------------------------------------------------------|
| cloud | 0.27 | 0.42 | 0.40 | +0.13 |
| free troposphere | 0.12 | 0.07 | <0.01 | -0.11 |



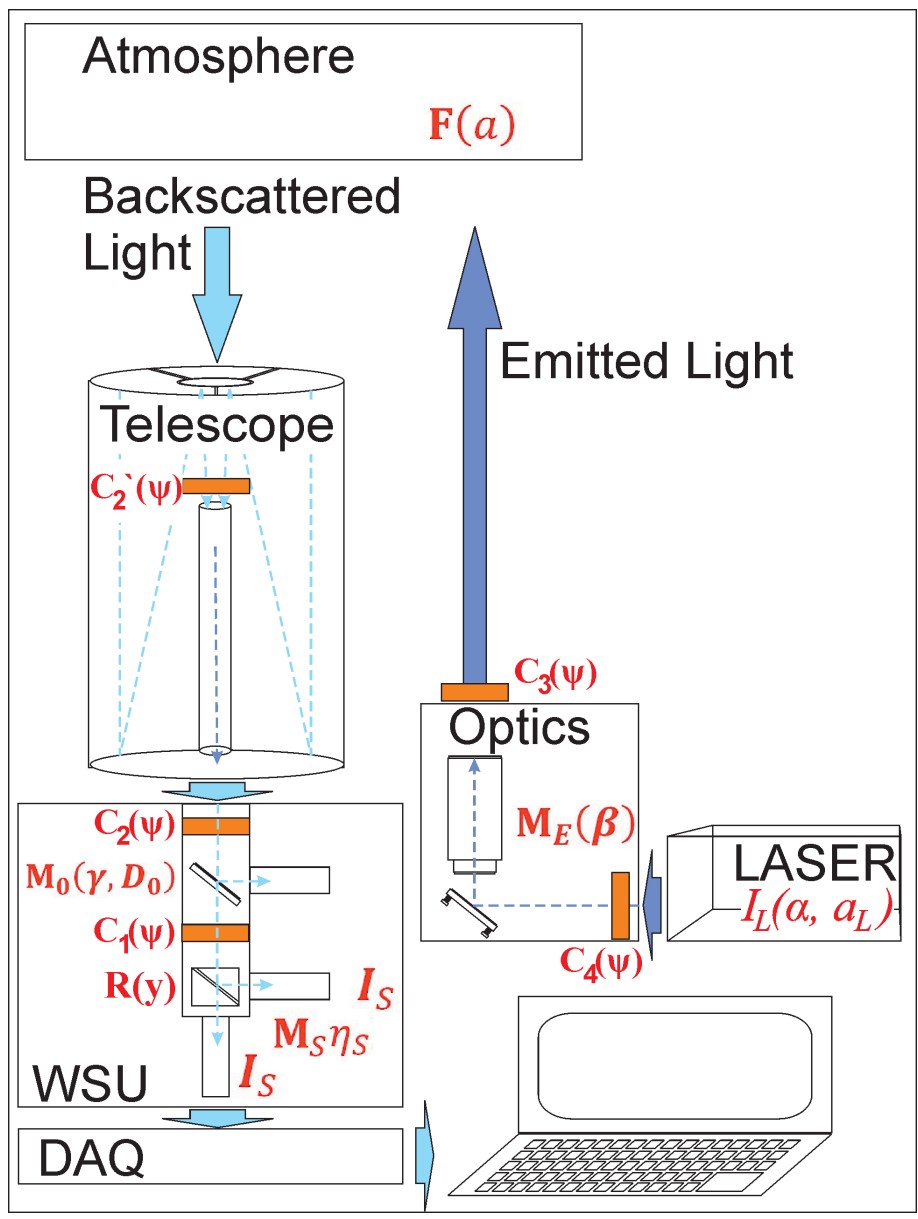

**Figure 1.** Schematics of a lidar system: emission block: laser and the emission optics; receiving block: telescope, wavelength separation unit (WSU) and data acquisition (DAQ) block. - Mueller-Stokes notations (red) for specific optical components used further in the study; - Possible positions of the calibration units (orange) in respect to the optical layout; DAQ: Data Acquisition



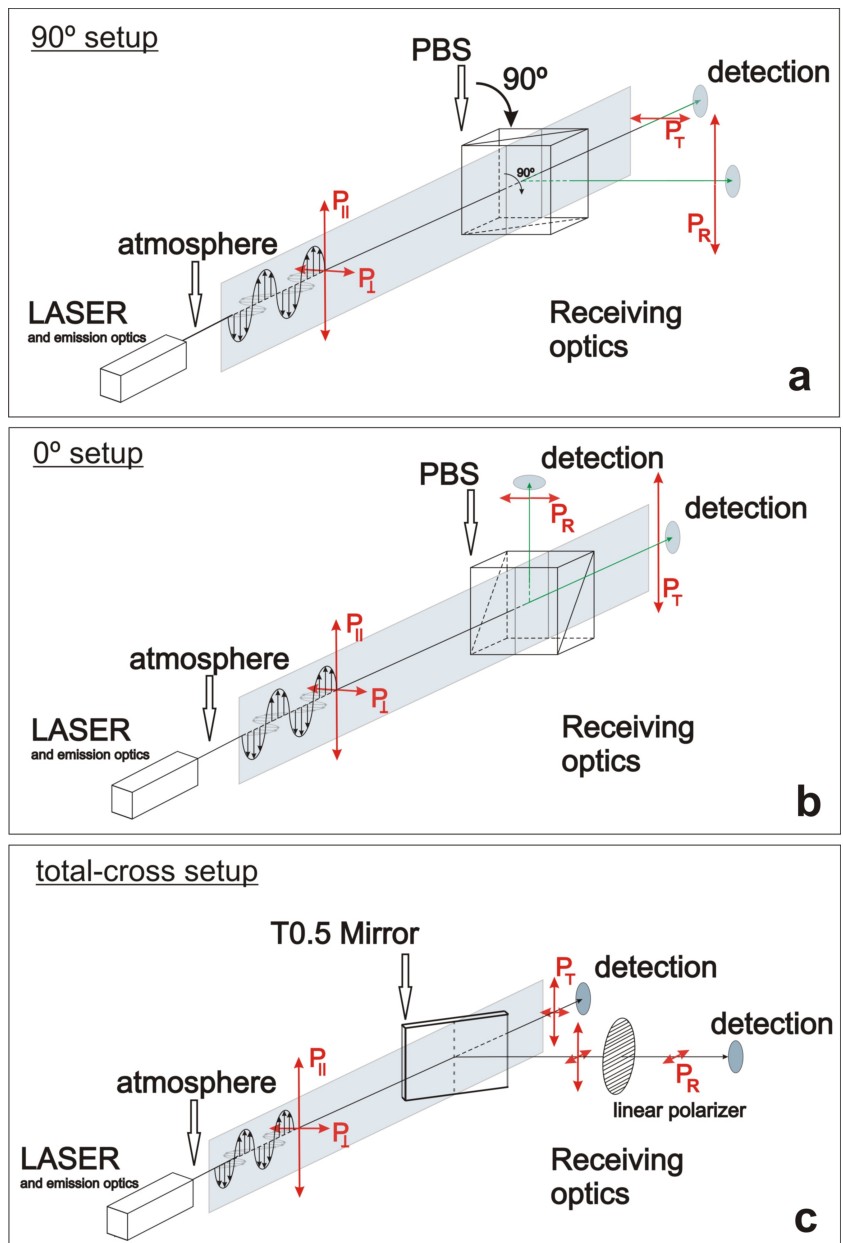

**Figure 2.** Detection setups (according with (Freudenthaler et al., 2009)). $P_\perp$ and $P_\parallel$ - collected radiation (parallel and cross components). $P_T$ and $P_R$ - detected components of the collected radiation (with contribution from the receiving optics). a) 90° detection setup; b) 0° detection setup; c) detection setup for PollyXT type lidar systems. PBS - Polarizing Beam Splitter, T0.5 mirror - 0.5 transmittance reflecting mirror





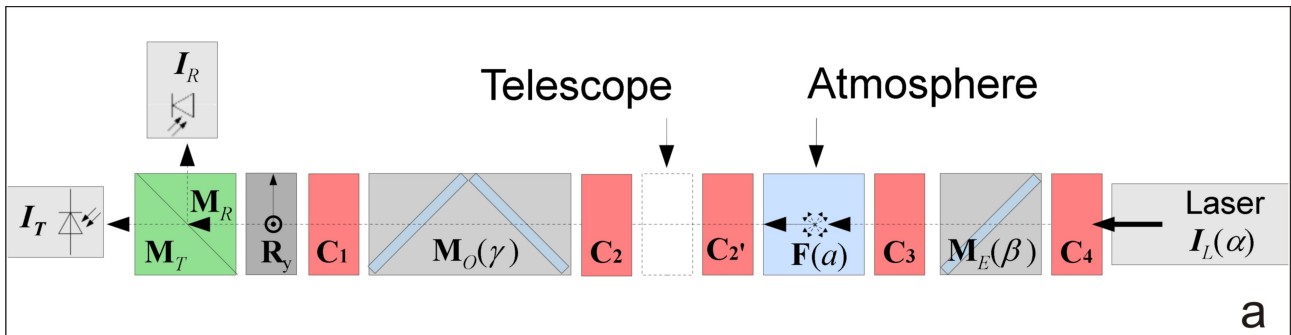

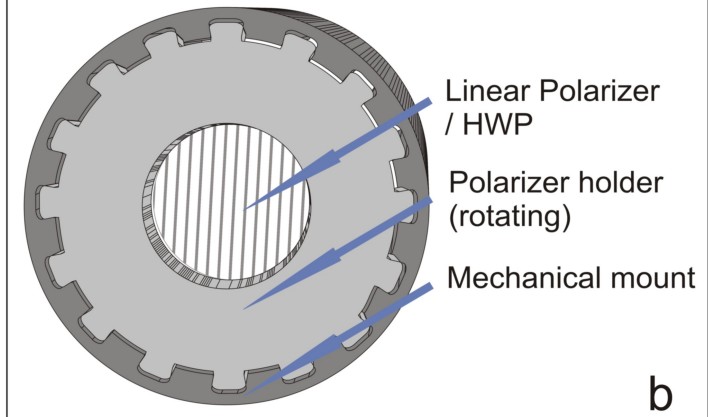

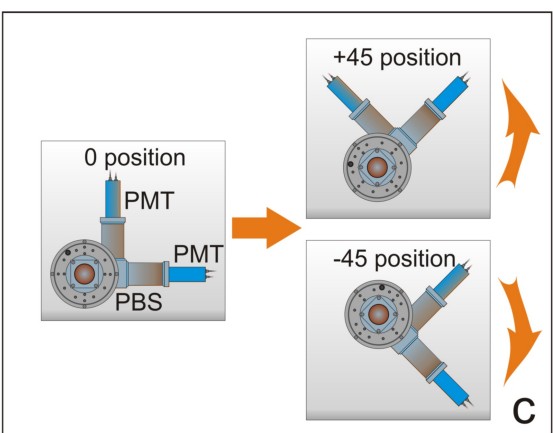

**Figure 3.** Calibration blocks: a) schematics of a polarization sensitive lidar system with Mueller matrix block elements, different calibrator positions (red blocks - C); b) multiangle mechanical rotator mount for the $\Delta 90°$ polarizer calibrator or the $\Delta 90°$ optical calibrator. 22.5° rotation step. HWP - Half Wave Plate; c) mechanical rotator for the $\Delta 90°$ mechanical calibrator. PMT - photomultiplier, PBS - Polarizing Beam Splitter



**Figure 4.** Numerical simulation: a) calibrated signal ratio - $\alpha$ simulations for different atmospheric depolarization values - $\alpha(0°:180°)$; b) calibrated signal ratio - $\alpha$ simulations for different atmospheric depolarization values - $\alpha(-10°:10°)$; c) $\eta$ - $\alpha$ simulations for different atmospheric depolarization values





**Figure 5.** Numerical simulation: a) Y - $\alpha$ simulations - for several atmospheric depolarization values; b) Y - $\alpha$ simulations - for several effective diattenuation values; c) Y - $\alpha$ simulations - for several effective diattenuation values - zoom in



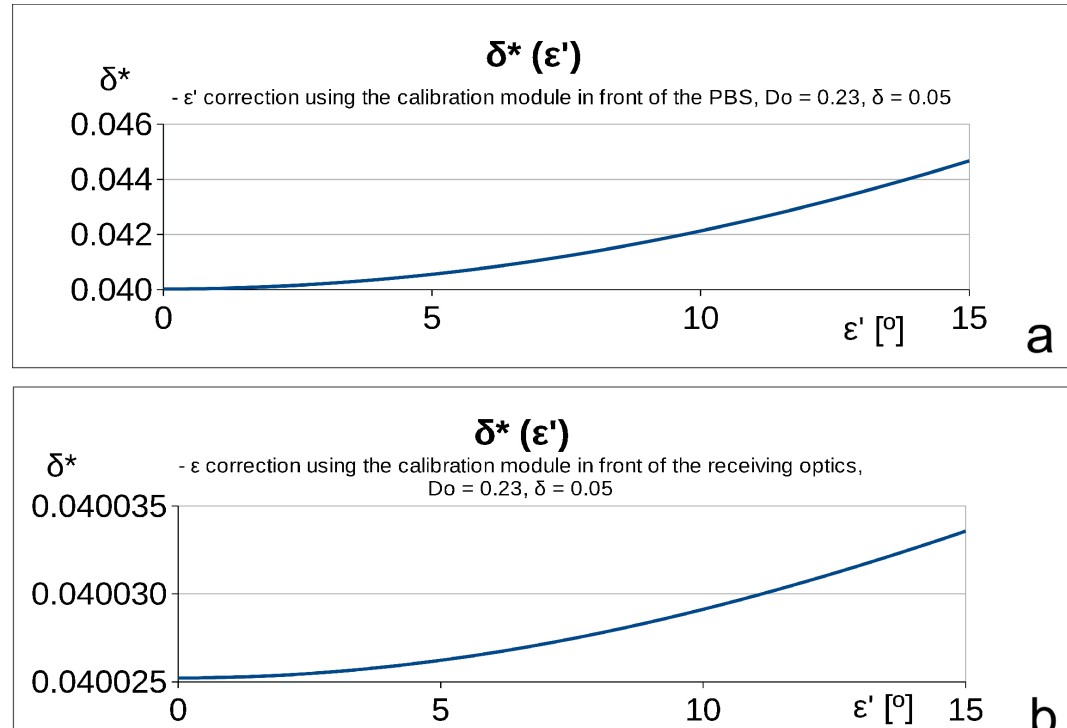

**Figure 6.** Numerical simulation: a) measured depolarization ratio - $\varepsilon'$ dependence when using the $\alpha$ correction module in front of the PBS b) measured depolarization ratio - $\varepsilon'$ dependence when using the $\alpha$ correction module in front of the receiving optics





**Figure 7.** Calibration values using two experimental techniques: polarizer rotation calibrator: $\eta^*_{before}$ and mechanical rotation calibrator: $\eta^*_{after}$ - a) for the Bucharest lidar; b) for the Granada lidar; c) assessment and correction of $\alpha(Y)$ parameter for the Bucharest lidar using the iterative procedure, mechanical rotator in front of the PBS





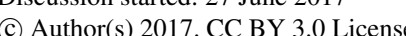

**Figure 8.** Data from $26^{th}$ of September 2013 at the RALI-Bucharest lidar system. a) range corrected time series at 532nm; b) volume linear depolarization ratios for not-corrected, $(\eta^*, a_L, D_O)$ corrected and $(\eta^*, a_L, D_O)$ and $\alpha(Y)$ corrected profile at 532nm - smoothed data, one hour average; c) particle depolarization ratio profile at 532nm - smoothed data, one hour average







**Figure 9.** Data from 12 of July 2012, at the Mulhacen-Granada lidar system a) range corrected time series at 532nm; b) volume linear depolarization ratio - corrected profile at 532nm - smoothed data, one hour average; c) particle linear depolarization ratio - corrected profile at 532nm - smoothed data, one hour average; d) HYSPLIT back-trajectories analysis (-144h) for the detected layers, at the Granada site





**Figure 10.** Data from $06^{th}$ of August 2012, at the MUSA-Potenza lidar system a) range corrected time series at 532nm; b) volume linear depolarization ratio - corrected profile at 532nm - smoothed data, one hour average; c) particle linear depolarization ratio - corrected profile at 532nm - smoothed data, one hour average; d) HYSPLIT back-trajectories analysis (-120h) for the detected layers, at the Potenza site



**Figure 11.** Data from $17^{th}$ of April 2010, at the Maisach-Munich lidar system a) range corrected time series at 532nm; b) volume linear depolarization ratio - corrected profile at 532nm - smoothed data, one hour average; c) particle linear depolarization ratio - corrected profile at 532nm - smoothed data, one hour average; d) HYSPLIT back-trajectories analysis (-72h) for the detected layers, at the Munich site