# Peer review of "Experimental techniques for the calibration of lidar depolarization channels in EARLINET"

_Atmospheric Measurement Techniques, 2017_

## Referee Comment (RC1) · Anonymous Referee #2 · 20 Jul 2017

The paper surveys different procedures for calibration of lidar systems to improve the accuracy of depolarization measurements, which are of primary importance to infer aerosol particle shape, hence typology. It briefly introduces the formalism to be used in light polarization measurements and reviews the current calibration techniques, describing relative merits and drawbacks. Then present results from the application of such calibration techniques on some case studies.

There are many paper dealing with the calibration of polarization diversity lidars, but the originality of the present one resides in its cut, more oriented to the description and practical implementation of the calibration systems, than in their theoretical description. This, in addition to a praiseworthy review of the theoretical assumptions of existing calibrations, assures it of the interest of the community, and for this reason I believe

that the work deserves the publication.

There are however a few issues the authors may want to further clarify, which I detail below, together with some minor items.

(2,8) typo "... about to..."

(2,20) "... distinguish between rather spherical particles with low depolarization ratio, and non-spherical particles with higher depolarization ratios." I would here drop the adverb "rather", as the whole sentence tends to suggest (at least to me) a univocal relationship between polarization and a "degree of asphericity", which is misleading, as it has been proved, for instance with theoretical T-matrix computations, that particles that are "rather" but not exactly spherical (i.e. prolate or oblate spheroid with aspect ratio close to unity) may have values of depolarization higher than considerably "more aspherical" (i.e. with aspect ratio much different than one) particles.

(2,22) "...low depolarizing (e. g. local aerosol) ...", well this claims depends on where you lidar is placed, that in turn dictates what is to be considered "local". I guess that a scientist working in Tamanrasset would have different views on what to consider "local". So you may consider to change "local", to "urban aerosol", as instance?

(4,19) A polarization purity of 95% is definitely a problem, and this should be stressed (actually is quite pessimistic, but even a more common 100:1 polarization purity still is a problem). Here you can quote that the residual non polarized laser light can be easily filtered out. It is said thereafter but I think the best place to pose that remark is here.

(7,12) please use "responsivities" instead of "quantum efficiencies", as the latter is only a factor of the former. This has an impact in what follows.

Formula (19): this is basically the ratio of the overall photodetector responsivities for the two channels. What follow is my crucial remark, and I would like the authors to discuss it in some more length. The responsivity, or the "gain" of a detector, is the ratio between the power input (in our case the photon flux) and output, (the current, or photoelectron

rate). One would like this gain to be constant, i.e. the idealized detector should have an output which is linearly related to the input. Unfortunately, this is seldom the case for the PMTs and APDs, as the gain may be dependent from the level of the input (this claim is straightforward in the photoncounting acquisition mode, due to "dead time" counting effects, but it is also true in current acquisition mode). This makes the ratio in (19) possibly dependent on the measurement conditions, i.e the altitude at which this ratio is computed (this is somehow implicitly – too much implicitly - addressed in fig 7 a-b) and, in some cases more important, on the level of sky background. It may well be that for "EARLINET-like" systems, which often use high power, low pulse repetition rate lasers, and very narrow interferential filters to reduce background to levels much lower than the actual signal, this effect is not apparent; but in general, and especially for systems with larger spectral bandwidth and low power, high pulse repetition rate lasers, this may be an issue. This is an issue which, to my knowledge, has never been addressed in any study putting forward the merit of calibrations others than the "0° calibration", and I think it is worthwhile to mention it.

(8,18-19) The sentence is unclear and should be rephrased.

(16,18-20) I am somehow uncomfortable with the whole sec. 4.4. I guess everyone is already well aware of "... the importance of calibrated depolarization lidar products...". This is not the main goal of the paper, but rather to discuss at length the different calibrations; hence it would have been much more interesting to show what is the effect of these calibration procedures, i.e. to show uncorrected vs corrected profiles, which I think is a display much more in line with the rest of the paper. Therefore, I would ask the authors to do that.

(16,24) I understand that the presentation of particle depolarization is functional to the aim of showing "... the importance of..." (see above), but again I think this is not the main message the paper is delivering. Moreover, as correctly stated, the computation of particle depolarization is affected by uncertainties on the aerosol backscatter coefficient and this is especially true in the case of low aerosol loading, as correctly stated in

(17, 14-16). This should open a completely new and wide discussion which is clearly beyond the scope of the article, which is on calibration procedures: the effect of these may be heavily masked by other sources of inaccuracies in the computation of particle depolarization.

In synthesis, I would ask to rewrite sec. 4.4, presenting calibrated and uncalibrated profiles on selected case studies, or to drop it entirely.

(18,24-25) This is a very nice result which I think is understated, as the comparison of the observations in regions supposedly free of aerosol, with the theoretical values of the molecular depolarization is the key factor to assess the goodness of the calibration procedures hereby described. The authors may consider to add a table reporting the values of "low aerosol height range values", vs the molecular depolarization as expected from theory. The bandwidth of the interferential filter should of course be also quoted, as it impacts that value. Incidentally, it might be quoted (18,27) that also the presence of small amount of liquid aerosol may impact the profile, in a different direction and to a smaller extent.

I think this is a very nice paper, and I would like to see it published.

---

## Referee Comment (RC2) · Anonymous Referee #3 · 21 Jul 2017

The paper explains calibration procedures for lidar depolarization measurements and compares and contrasts various methods used in EARLINET. This will be a very useful reference for EARLINET operators and for those wishing to understand the data quality of EARLINET depolarization measurements. I would like to see it published. However, the manuscript suffers somewhat from a sub-optimal organization related (perhaps) to a confusion about its primary purpose.

I have two major concerns. First, I spent hours just trying to understand the paper. This included a lot of time paging back and forth to find variable definitions. There are 16 variables in the first equation which are explained in a somewhat scattered way in the following paragraph, and ultimately the final equation of the theory section includes 25 variables, only some of which are the same as in Eq. (1). This is many more variables

than I can keep in my head at once. Later sections refer to quantities only by variable name without any verbal description or reminder of what the variable means as if we have everything memorized. It was good to see the list of variables at the end, but this is not sufficient. If this paper is going to be useful for its recommendations or as a reference for data quality, it should be written clearly and consicely for a target audience who will probably want to use it practically, not theoretically. While it's admirable to see the theory treated in such a thorough way, I'm not sure there is anything new in the theory section. Rather, I think the purpose of this section should be to lay the foundation for understanding the calibration procedures and results that will be discussed in later sections. To that end, is it possible to streamline the derivations and to present the equations in a simplified way such that they clarify the relationships between the quantity you would like to assess (a or delta), the quantities that are more directly measured (Stokes vectors) and the calibration parameters that are going to be discussed (diattenuation parameters and offset angles), without every detail of scattering theory being included? I have to admit that ultimately I failed to thoroughly understand the theory section although I am familiar with these concepts using different equations and different variable names and symbols. So possibly I'm wrong and all this really is indeed needed. In that case, it is even more important to make this section pedagogically clear. Describe in words the purpose of each part of the derivation, end sections with the most simplified useful version of the equations (like the equations that undergraduate textbooks enclose in a box), and restate the variable meaning and not just the symbol each time a variable is reintroduced in a later section. You'll need to write it as if you are teaching it, not just demonstrating that you know it well yourself.

My other concern is about the results section. You have stated two purposes, given at the start of section 4.4: to present the importance of calibrated depolarization products and to assess the accuracy of the calibrated depolarization products. I suggest that the first purpose is misplaced here. Except for the brief discussion in the introduction that can be seen as motivation, this paper doesn't need to show the importance of calibrated products. The second objective, to assess them, is of far more importance,

and there is room for improvement in how this objective is addressed. It's good that you have examples to show that the measured depolarization is close to the expected value, especially for the aerosol-free molecular depolarization which is known independently. This should be expanded. Is there any other simultaneous data available for independent assessment or inter-comparison? Other than these comparisons, you also have error bars which can give an idea of the precision of the depolarization measurements. Please be more thorough in explaining how the error bars are calculated and make sure they are consistent in the various comparisons, because these are a large part of the rather small set of information available to assess the results of the calibrations presented.

Specific comments:

Page 1, line 4. Which "derived parameters". Please be specific in the abstract.

Page 2, line 18. What are "all relevant parameters". I think "all relevant parameters are shape dependent" might be a bit of an overstatement, but the rest of the paragraph does a good job of explaining when depolarization measurements need to be highly accurate and when they are used just in a relative sense.

Page 2, lines 25-30. These two sentences should be rewritten to make your point more clear. What do you mean by "ranges around close values", that the depolarization values are clustered well so that different types are distinguished easily, or the opposite, that different types have similar values and can't be distinguished unless the depolarization is very accurate? What does "The same issue" refer to?

Page 3, line 27 - Page 4, line 11. With so many variables, it would be helpful to organize the descriptions more predictably. Please either describe all the variables left to right, including the dependent variables, or else describe all the dependent variables and then all the independent variables. Or simplify as discussed in the general comments above, and then maybe fewer variables will be needed.

СЗ

page 4, line 22. A new variable  $\varepsilon$  is introduced without being explained.

page 7, lines 2-9. "For most cases we consider" suggests there is a much simpler version of the equation that is being used for the rest of the paper. Please give this simpler version explicitly.

page 7, Eqn 20. Is the variable  $\eta$  the same as the sub-scripted variable  $\eta_s$  from Eqn 1?

page 9, line 11. Spell out acronyms, Half Wave Plate

page 13, line 11. Reintroduction of variable Y after 4 pages. Here is an example where it would be easier to follow if you remind readers what variable Y means and where it was introduced, something like "Y, which was introduced in Eq. (24) and is mathematically related to the error angle". Or better yet, since the error angle is a more familiar variable than Y, maybe consider recasting the plots in Fig 5 to use error angle instead of Y.

page 13, line 22-27. Here is the first time where you make it explicit that correcting errors in the angle with hardware is better than post-processing. I found it very confusing before this part of the paper. While I understand that this paper aims to treat all methods of calibration used in the various EARLINET instruments, the earlier discussion of the two methods (that is, hardware correction and analytical correction in post-processing) did not make a clear distinction between them and I was left wondering if for some strange reason the authors were only considering the post-processing solution, which is the less desireable one. Please do everything you can do to make all options clear from the start and to be certain to distinguish clearly between calibration methods that change (and therefore correct) the angle errors from methods that do not change them (and therefore have to mathematically adjust the results in post-processing) at every stage of the discussion. Don't leave any mysteries to be solved at the end of the paper.

page 17, Conclusion section. This section is good. It is very helpful that the calibration methods are summarized again here because it helps to clear up some of the confusions from earlier in the paper.

Page 18, line 20. "Proper corrections". Please make this sentence more specific. Are you talking about only the diattenuation correction here, or also about the angle corrections?

Page 18, lines 22-30. This discussion is critical to your assessment of the calibration results. It is out of place appearing for the first time in the conclusions. This should be part of the results and discussion, and it should be expanded.

References: the last 3 references are out of order.

Figure 4b. I'm confused about why the true and measured depolarization values don't agree even at an angle error of zero. Is this because there are other calibrations that have not been applied? Given that the point is to show the effect of angle error on the depolarization, then I think the angle error should be the only uncorrected error in the simulation.

Figure 4 caption. There is a typo. The range of alpha is 0 to 10 degrees, not -10 to 10.

Figures 8,9,10,11. What do the error bars represent (systematic or random, empirically calculated from data variability or theoretically calculated)? Please explain in the figure caption and in the text.

Figure 9. Why is there no depolarization data below 2000 m?

Figures 8,9,10,11. Please make all the y-axis lower limits the same for all the subpanels in a given figure.

---

## Referee Comment (RC3) · Anonymous Referee #1 · 22 Aug 2017

The depolarization ratio is an important property to characterize different aerosol types. An exact measurement of this quantity is therefore an important issue. The manuscript describes a method to assess and to correct for the diattenuation of receiving optics and the rotation angle between the laser and the receiver. Both quantities can be determined by the use of the Delta 90° calibration, which is an important method for the proper characterization of a lidar system. The description (figures and formulas) needs to be improved. It is shown that these corrections lead to a significantly better result for the volume depolarization ratio (Fig 8). The lidar community will profit from these techniques. Therefore I recommend it for publication, although there are some mayor points that have to be improved:

Major Remarks

1. Equation (4) is wrong. If you multiply two vectors you'll get a scalar. The first vector (i\_E, q\_E, ...) seems to be I\_E, but should be the matrix M\_E.

2. p6, I14 How do you assure that there is no misalignment of the additional polarization filters after the PBS? Or which error would a misalignment by  $1^{\circ}$  introduce to the depolarization? Please comment on this.

3. p8, l18 What do mean by "all effects"? Be more precise.

4. p8, I30 – p9, I8 I am not sure whether it is necessary to introduce the "45° calibration". The problems with this calibration are already discussed in Freudenthaler et al., 2009. Just keep the focus on the Delta 90° calibration. In the following text I would recommend to skip the quotation marks " " around the Delta 90° calibration.

5. p9, l26 You mention a set of two relative Delta  $90^{\circ}$  calibrations, but the equations (24) and (25) make use of only one calibration measurement.

6. p10, I5-9 This is a very important and interesting point. You will use only one calibration method (the Delta 90° calibration) to assess different parameters of your lidar system (at different positions in the system). It would be good to state this clearly already in the introduction, because this could be something like a red line through your paper. For persons not so experienced in the lidar business it would help to start with the calibration effort.

7. p10 chap. 3.2.1 A HWP calibrator works for single wavelength lidar systems only. In multiple wavelength lidar systems a HWP must be placed in front of each PBS, but not in front of the receiving optics. This should be mentioned.

8. p12, eq (26) and eq (27): Do you mean  $D_O$  ("O like orange") or  $D_0$  (0 – zero)? You use both notations throughout the manuscript, not only in these two equations. Please choose one notation for the entire manuscript.

9. p13, l1, l4 What do you mean by "retrieved measured" and "simulated measured" ? It is simulated or measured? Maybe something like "simulated apparent calibration fac-
tor" (see Freudenthaler, 2016) and "retrieved calibrated signal ratio" as you call delta\* in Fig 4. Please use one name for a certain quantity throughout the entire manuscript.

10. p13, I23 and Fig 5c How do you get the uncertainty of 25%? For the known value of alpha =  $10^{\circ}$  and D\_0 = 0.25, Y would be -0.45. You have to assume some uncertainty for alpha and D\_0, which you have not mentioned.

11. p13, chap 3.4.2 You discuss the possibility of correcting alpha with a HWP or mechanical rotator in front of the PBS. At this point you should discuss the possibility of turning the linear polarizer in front of the PMT (setup 2c) to correct for alpha. It should lead to similar results as the rotation of the PBS.

12. p15, I17 Why you use alpha(Y)? Is there a reason for the dependence on Y? Please explain it or change it. This holds for the following text.

13. p15, I31 and Table 3 You mention a correction of a\_L. Please indicate the value and its uncertainty for a\_L.

14. Fig 8 The color scale for the time-height plot looks unorganized and has no description. Why is there a grey line between the color plot and the color scale? The same holds for Fig 9a, 10a, 11a.

15. For all Fig 8 - 11 Please indicate the vertical smoothing length for your profiles.

16. p16, l25-30 and Fig 9 The Granada measurement is presented in unorganized way. I would recommend to skip it as you have already an example for Saharan dust over Potenza or to take into consideration the following comments:

- Every plot starts at a different height.

- Up to 7 or 8 km height would be sufficient.
- Heights are about ground or sea level?
- Why the back trajectories are calculated for so close height ranges? Why at 2000
UTC and not 2100 UTC?

- "One hour average", the red lines in Fig 9a show only 50 minutes.

- p16, I26-27 If the particle linear depolarization ratio is close to the molecular depolarization in the low aerosol height ranges, then something is wrong. The particle linear depolarization ratio depends on the aerosol type, if there are only few aerosols, the particle linear depolarization ratio gets noisy but not close to the molecular depolarization.

- 0.22 (two significant digits are enough) is quite low for mineral dust. Please compare it to literature values. It could be polluted or mixed dust.

- "several days" How many?

17. p16, I31 – p17, I3 and Fig 10 The Potenza case is better organized. Again: The plots up to 7 or 8 km would enlarge the interesting part (for example the Munich case is shown up to 5 km to focus on the interesting part). Height about ground level or about sea level is important for a mountain station like Potenza. The shown measurement starts on 6th August 2012 00:00 UTC, the profiles indicate 05.08.2012. The abbreviation PBL is not explained.

18. p17, I23 and Fig. 5 What do you mean by "effective diattenuation"? Where is the difference to "diattenuation" (of the receiving optics)?

19. p17, I33 – p18, I4 The HWP might be the best solution for single wavelength lidar systems. Please discuss the use of dual wavelength polarization measurements as well.

20. p18, I5-10 Not only here, but in the whole conclusion: Please do not forget to discuss the calibration by a linear polarizer.

21. Appendix, p19 How are your angles alpha, beta and gamma defined? As rotation around the propagation axis or with respect to the PBS? p15, I17 you state: "the ro-
tation of the plane of polarization of the laser with respect to the PBS: alpha(Y)." The axis of propagation and all angles around it are defined by the orientation of the PBS. Please make a clear statement when introducing the angles.

22. The cited literature seems to be at the situation of 2015 where the paper was submitted for the first time (except of the accompanying papers by Bravo-Aranda and Freudenthaler, both AMT 2016). Please update the reference list to include more recent publications.

23. Table 1: If we can rotate the emitted light with a HWP to  $+-45^{\circ}$ , it should also be possible to mechanically rotate the emission unit to  $+-45^{\circ}$  leading to the same result. A linear polarizer could also be placed in the emitter unit as described in Chapter 8.3 in Freudenthaler, AMT 2016. Please add these two possibilities in the column "position". A column for single or multiple wavelength use could be added, as the mechanical rotator and the polarizer rotator can be used for several wavelengths, whereas the HWP can be used for a specific wavelength only.

24. Table 2: How do you explain the differences between your Table 2 and Table 5 in Bravo-Aranda et al., AMT 2016 regarding the Munich lidar system? You report  $D_0 = 0.059$  (at 532 nm ?), the other publication reports  $D_0 = 0.011$  at 532 nm. Or is it valid for different years? The Potenza lidar system differs by the sign only,  $D_0 = +0.055$  (this manuscript),  $D_0 = -0.055$  (Bravo-Aranda, 2016).

25. Table 3: Please specify the height range (for "cloud" and "free troposphere") used for your average.

26. I dare that the figures (Fig 4-11) do not fulfill the standard of the journal. A professional plot program should be used.

Minor Comments:

- The Spanish institute affiliations are without street name, while all other institutes are located in a certain street and number.
- There should be a space between the number and the unit, for example 3.5 km and 532 nm. Please go through your entire manuscript to check this.

- p3,l2 methods (add plural s)

- p3, l14 "(\alpha)" is not necessary in the introduction.

- p4, I20-21 Change to normal fond (not italic, not bold).

- p6, l1 "All optical elements M\_O can be described by Mueller matrices of diattenuators M\_D with retardation M\_ret" and rotation M\_gamma. As you show in the next line (equation (8)).

- p7, eq (15) Please consider the rules for notation as you stated correctly on p3, l27 "bold italic fonts are used for the Stokes vectors, bold for the Mueller matrices and italic for the scalar variables". The same holds for  $G_S$  and  $H_S$  in eq (16) and (17).

- p7, I8 Use capital E as index.  $u_E = sin (2 alpha) * a_L$

- p7, eq (21) Please use capital R and T.

- p8, I23 "This method" To which of the two mentioned methods you are referring. Be more precise.

- p8, I31 "the larger is the error ... "

- p9, l21-22 "Table 1 summarizes main advantages and disadvantages when using different calibration techniques for the Delta 90° calibration." Put this sentence a little earlier (p9, l18), before you start describing how to find the zero degree position.

- p9 eq (24) What is eta\*\_pol ? It is not introduced. eta\* would be sufficient.

- p10, l17 polarizing beam splitter

- p11, I5 and Fig 3 "optical rotator calibrator" please do not use different words for the same thing throughout your manuscript. "HWP calibrator" or "optical rotator calibrator" ?
- p11, I29 use \citet{} instead of \citep{} command.

- p12, l3-4 "measuring two depolarization channels at 532nm and a 90° setup" Reshape this part, the lidar does not measure a channel nor a setup.

- p12-13, chap. 3.4 You are talking about the depolarization ratio, not the polarization ratio. It occurs 5 times in this chapter the wrong expression.

- p13, l11-12 The polarization parameter (a) is not the atmospheric depolarization that you show in your figure.

- p13, l12 "the diattenuation parameter" of the receiving optics.

- p13, l22 "The analytical correction of alpha can be performed by" determining G\_S, H\_S and K "using Eq. (16), (17) and (21)."

- p15, I2-3 Already said on p12, I11-12
- p15, l16 polarization

- p15, l19-21 "we will only consider the post measurement analytical correction" and one line later you start with the experimental correction for alpha and Fig 7c. For the post measurement analytical correction, did you use alpha =  $10^{\circ}$  or alpha =- $0.04^{\circ}$ ? Please state this clearly at the beginning of the discussion about the post measurement analytical correction.

- p15, l31 "lower values" Give numbers.
- p15, I33 Put the citation in one bracket.
- p16, l19 "use the same"
- p17, I22 its placement ... keep normal text fond

- p17, l27-28 "while the method[s] that use the calibrator in front of the PBS allow[s] to take into account ..." Plural or singular?

**AMTD**
- p18, l10 "number of lidar systems it can be applied to" You mean single wavelength lidar systems or what are the other limiting factors?

- p18, I20, I23 Do not use "+" in the text. Use "and".

- p18, l26 0.01 - 0.03

- Appendix A: D has to be italic. M\_S has to be written with a capital S.

- Keep the alphabetical order of your reference list (e.g. Reichardt should not appear after Winkler, page 25)

- When citing make sure, that David. G, et al, 2012 appears as David et al, 2012

- Table 3: The index "pol" in eta\*\_pol is not used elsewhere in the case study. In Fig 8 it is just called eta\*.

- Fig 4 Please add in the caption "delta\*" b)  $(0^\circ : 10^\circ)$  In my opinion the eta\* plot (Fig 4c) is not useful to show. Or it has to be discussed in more detail. What are the implications for the parameter K?

- Fig 5 It would be better to show in Fig 5 a+b only the range from  $0^{\circ}$  to  $20^{\circ}$ .

- Fig 6 In the caption: "measured depolarization ratio" – "calibrated signal ratio delta\*" would be better. Fig 6b in the title of the diagram the ' is missing for epsilon.

- Fig 8 In the caption "one hour average" – the red lines indicate just 45 minutes. What is correct?

References: Bravo-Aranda, J. A., Belegante, L., Freudenthaler, V., Alados-Arboledas, L., Nicolae, D., Granados-Muñoz, M. J., Guerrero-Rascado, J. L., Amodeo, A., D'Amico, G., Engelmann, R., Pappalardo, G., Kokkalis, P., Mamouri, R., Papayannis, A., Navas-Guzmán, F., Olmo, F. J., Wandinger, U., Amato, F. and Haeffelin, M.: Assessment of lidar depolarization uncertainty by means of a polarimetric lidar simulator, Atmos. Meas. Tech., 9(10), 4935–4953, doi:10.5194/amt-9-4935-2016, 2016.

AMTD
Freudenthaler, V., Esselborn, M., Wiegner, M., Heese, B., Tesche, M., Ansmann, A., Muller, D., Althausen, D., Wirth, M., Andreas, F. I. X., Ehret, G., Knippertz, P., Toledano, C., Gasteiger, J., Garhammer, M., and Seefeldner, M.: Depolarization ratio profiling at several 10 wavelengths in pure saharan dust during SAMUM 2006, Tellus B, 61(1), 165–179, 2009.

Freudenthaler, V.: About the effects of polarising optics on lidar signals and the 90 calibration, Atmos. Meas. Tech., 9(9), 4181–4255, doi:10.5194/amt-9-4181-2016, 2016.

**AMTD**

---

## Author Comment (AC1) · 27 Sep 2017

First we would like to thank Referee #1 for the time dedicated to provide the comments and suggestions for the interactive discussion. The comments had definitely improved the manuscript clarity and overall presentation. We hope that the updated version is satisfactory. Below you will find a point by point description of how each comment and suggestion was addressed. Please see the attached pdf file for the detailed answer

Please also note the supplement to this comment:
https://www.atmos-meas-tech-discuss.net/amt-2017-141/amt-2017-141-AC1-supplement.pdf

---

## Author Comment (AC2) · 27 Sep 2017

We would like to thank Referee #2 for the valuable contribution to this manuscript and for the time dedicated to providing the comments and suggestions for the interactive discussion. We made all efforts to follow all recommendations and we hope that the updated version is satisfactory for publication. Attached you will find apoint by point description of how each comment and suggestion was addressed.

Please also note the supplement to this comment:
https://www.atmos-meas-tech-discuss.net/amt-2017-141/amt-2017-141-AC2-supplement.pdf

---

## Author Comment (AC3) · 27 Sep 2017

First we would like to thank Referee #3 for the time dedicated to provide the comments and suggestions for the interactive discussion. We made all efforts to follow all recommendations provided by the referee. We hope that the updated version is satisfactory. Attached you will find a point by point description of how each comment and suggestion was addressed.

Please also note the supplement to this comment:
https://www.atmos-meas-tech-discuss.net/amt-2017-141/amt-2017-141-AC3-supplement.pdf

---

## Author Comment (AC4) · 27 Sep 2017

Please find attached the updated version of the manuscript "Experimental techniques for the calibration of lidar depolarization channels in EARLINET"

Please also note the supplement to this comment:
https://www.atmos-meas-tech-discuss.net/amt-2017-141/amt-2017-141-AC4-supplement.zip

---

## Author Comment (AC5) · 28 Sep 2017

The second update includes several corrections to the bibliography

Please also note the supplement to this comment:
https://www.atmos-meas-tech-discuss.net/amt-2017-141/amt-2017-141-AC5-supplement.zip

---

## Author Response (AR2)

Anonymous Referee #1 Comments:

The manuscript has improved a lot since the first submission.
There are still three points to consider:
1.) A HWP used to calibrate or to correct for alpha works only, if the depolarization measurement is performed at a single wavelength. A HWP for 532 would be a quarter wave plate for 1064 and introduce circular (or elliptical) polarization. Dual-wavelength polarization lidars need a HWP for each wavelength to perform a calibration.
2.) The change from 0 (zero) to O like Optics has been done everywhere except of Equation (17). Please change it.
3.) p16, l26-27 You mix particle and volume linear depolarization ratio. The first part of the sentence describes the particle linear depolarization ratio (which is around 0.22, see Fig. 9), whereas the second part of the sentence describes the volume depolarization ratio, which is close to the molecular depolarization ratio.

First of all I would like to thank Anonymous Referee #1 for time dedicated to provide the comments and suggestions for the manuscript. We followed all recommendations and we hope that the updated version is satisfactory for publication.

Comment 1: A HWP used to calibrate or to correct for alpha works only, if the depolarization measurement is performed at a single wavelength. A HWP for 532 would be a quarter wave plate for 1064 and introduce circular (or elliptical) polarization. Dual-wavelength polarization lidars need a HWP for each wavelength to perform a calibration.

Answer: The text was updated according to the suggestions.

3.2.1 $\Delta 90°$ mechanical rotation calibrator and HWP calibrator

The same type of calibrator can be also implemented by using a stepping motor rotation mount or a HWP mount which is placed in a holder with fixed and accurate positions at 0 and p/m 45 (or multiple positions) (Fig 3.b). Dual-wavelength polarization lidars need a HWP for each wavelength to perform the calibration (a HWP for 532 would be a quarter wave plate for 1064 and introduce circular or elliptical polarization).

3.4.2 Correction for the α parameter

The experimental correction of (Y) can be performed either by rotating the PBS in the WSU (without or together with the receiving optics) or by rotating the plane of polarization of the collected light using a HWP placed in front of the PBS or in front of the receiving optics (in the case of one wavelength lidar instruments or systems with separate optics for the depolarization channels since a HWP used to correct for alpha works only if the depolarization measurement is performed at a single wavelength).

Comment 2: The change from 0 (zero) to O like Optics has been done everywhere except of Equation (17). Please change it.

Answer: The equation was updated. Thank you for pointing this error.

Comment 3: You mix particle and volume linear depolarization ratio. The first part of the sentence describes the particle linear depolarization ratio (which is around 0.22, see Fig. 9), whereas the second part of the sentence describes the volume depolarization ratio, which is close to the molecular depolarization ratio.

Answer: Thank you for pointing this mistake. We had updated the text accordingly:

Updated text:

Measurements performed using the Granada lidar system (Mulhacen) in July 2012 show the presence of a distinct layer between 2.5 and 5 km (Fig 9.a-c). The volume linear depolarization ratio shows high values in the aerosol layer (0.18) and levels close to the molecular depolarization in the low aerosol height ranges. The particle depolarization ratio shows values reaching 0.22 in the layer. The back trajectories model indicates that the corresponding air mass originates in Northern Sahara